# A latent clinical-anatomical dimension relating metabolic syndrome to brain structure and cognition

Marvin Petersen[1]*, Felix Hoffstaedter[2,3], Felix L Nägele[1], Carola Mayer[1], Maximilian Schell[1], D Leander Rimmele[1], Birgit-Christiane Zyriax[4], Tanja Zeller[5,6,7], Simone Kühn[8], Jürgen Gallinat[8], Jens Fiehler[9], Raphael Twerenbold[5,6,7,10], Amir Omidvarnia[2,3], Kaustubh R Patil[2,3], Simon B Eickhoff[2,3], Goetz Thomalla[1], Bastian Cheng[1]

[1]Department of Neurology, University Medical Center Hamburg-Eppendorf, Hamburg, Germany; [2]Institute for Systems Neuroscience, Medical Faculty, Heinrich-Heine University Düsseldorf, Düsseldorf, Germany; [3]Institute of Neuroscience and Medicine, Brain and Behaviour (INM-7), Research Center Jülich, Jülich, Germany; [4]Midwifery Science-Health Services Research and Prevention, Institute for Health Services Research in Dermatology and Nursing (IVDP), University Medical Center Hamburg-Eppendorf, Hamburg, Germany; [5]Department of Cardiology, University Heart and Vascular Center, Hamburg, Germany; [6]German Center for Cardiovascular Research (DZHK), partner site Hamburg/Kiel/Luebeck, Hamburg, Germany; [7]University Center of Cardiovascular Science, University Heart and Vascular Center, Hamburg, Germany; [8]Department of Psychiatry and Psychotherapy, University Medical Center Hamburg-Eppendorf, Hamburg, Germany; [9]Department of Diagnostic and Interventional Neuroradiology, University Medical Center Hamburg-Eppendorf, Hamburg, Germany; [10]Epidemiological Study Center, University Medical Center Hamburg-Eppendorf, Hamburg, Germany

*For correspondence:
mar.petersen@uke.de

**Abstract** The link between metabolic syndrome (MetS) and neurodegenerative as well as cerebrovascular conditions holds substantial implications for brain health in at-risk populations. This study elucidates the complex relationship between MetS and brain health by conducting a comprehensive examination of cardiometabolic risk factors, brain morphology, and cognitive function in 40,087 individuals. Multivariate, data-driven statistics identified a latent dimension linking more severe MetS to widespread brain morphological abnormalities, accounting for up to 71% of shared variance in the data. This dimension was replicable across sub-samples. In a mediation analysis, we could demonstrate that MetS-related brain morphological abnormalities mediated the link between MetS severity and cognitive performance in multiple domains. Employing imaging transcriptomics and connectomics, our results also suggest that MetS-related morphological abnormalities are linked to the regional cellular composition and macroscopic brain network organization. By leveraging extensive, multi-domain data combined with a dimensional stratification approach, our analysis provides profound insights into the association of MetS and brain health. These findings can inform effective therapeutic and risk mitigation strategies aimed at maintaining brain integrity.

## eLife assessment

This **important** work contributes to our understanding of the combined effects of metabolic syndrome on fronto-temporal gray matter morphology from two large-scale datasets. The evidence

based on state-of-the art multivariate imaging analysis and detailed micro- and macrostructural contextualization analyses is **convincing** and provides an understanding of the neurological correlates of metabolic syndrome, although the study would have benefitted from the inclusion of longitudinal data.

## Introduction

Metabolic syndrome (MetS) represents a cluster of cardiometabolic risk factors, including abdominal obesity, arterial hypertension, dyslipidemia, and insulin resistance (*Alberti et al., 2006*). With a prevalence of 23–35% in Western societies, it poses a considerable health challenge, promoting neurodegenerative, and cerebrovascular diseases such as cognitive decline, dementia, and stroke (*Aguilar et al., 2015*; *Scuteri et al., 2015*; *Beltrán-Sánchez et al., 2013*; *Boden-Albala et al., 2008*; *Atti et al., 2019*). As lifestyle and pharmacological interventions can modify the trajectory of MetS, advancing our understanding of its pathophysiological effects on brain structure and function as potential mediators of MetS-related neurological diseases is crucial to inform and motivate risk reduction strategies (*Eckel et al., 2005*).

Magnetic resonance imaging (MRI) is a powerful non-invasive tool for examining the intricacies of neurological conditions in vivo. Among studies exploring MetS and brain structure, one of the most consistent findings has been alterations in cortical gray matter morphology (*Yates et al., 2012*). Still, our understanding of the relationship between MetS and brain structure is constrained by several factors. To date, there have been only a few studies on MetS effects on gray matter integrity that are well-powered (*Beyer et al., 2019*; *Lu et al., 2021*; *Wolf et al., 2016*; *Tiehuis et al., 2014*). The majority of analyses are based on small sample sizes and report effects only on global measures of brain morphology or a priori-defined regions of interest, limiting their scope (*Tiehuis et al., 2014*; *McIntosh et al., 2017*; *Sala et al., 2014*). As a result, reported effects are heterogeneous and most likely difficult to reproduce (*Marek et al., 2022*). Existing large-scale analyses on the isolated effects of individual risk factors (such as hypertension or obesity) do not account for the high covariance of MetS components driven by interacting pathophysiological effects, which may prevent them from capturing the whole picture of MetS as a risk factor composite (*Hamer and Batty, 2019*; *Opel et al., 2021*; *Schaare et al., 2019*; *Borshchev et al., 2019*). In addition, analyses addressing the complex interrelationship of MetS, brain structure, and cognitive functioning by investigating them in conjunction are scarce (*Yates et al., 2012*). Lastly, while previous studies adopted a case-control design treating MetS as a broad diagnostic category (*Lu et al., 2021*; *Wolf et al., 2016*; *Tiehuis et al., 2014*), a dimensional approach viewing MetS as a continuum could offer a more nuanced representation of the multivariate, continuous nature of the risk factor composite.

Despite reports on MetS effects on brain structure, the determinants and spatial effect patterns remain unclear. A growing body of evidence shows that spatial patterns of brain pathology are shaped by multi-scale neurobiological processes, ranging from the cellular level to regional dynamics to large-scale brain networks (*Fornito et al., 2015*). Accordingly, disease effects can not only be driven by local properties, when local patterns of tissue composition predispose individual regions to pathology, but also by topological properties of structural and functional brain networks (*Fornito et al., 2015*; *Seidlitz et al., 2020*). Guided by these concepts, multi-modal and multi-scale analysis approaches could advance our understanding of the mechanisms influencing MetS effects on brain morphology.

We argue that further research leveraging extensive clinical and brain imaging data is required to explore MetS effects on brain morphology. These examinations should integrate (1) a research methodology that strikes a balance between resolving the multivariate connection of MetS and brain structure while accounting for the high covariance of MetS components; (2) the recognition of impaired cognitive function as a pertinent consequence of MetS; and (3) the analysis of the spatial effect pattern of MetS and its possible determinants.

To meet these research needs, we investigated cortical thickness and subcortical volumetric measurements in a pooled sample of two large-scale population-based cohorts from the UK Biobank (UKB) and Hamburg City Health Study (HCHS) comprising in total 40,087 participants. Partial least squares correlation analysis (PLS) was employed to characterize MetS effects on regional brain morphology. PLS is especially suitable for this research task as it identifies overarching latent relationships by establishing a data-driven multivariate mapping between MetS components and brain

morphometric indices. Furthermore, capitalizing on the cognitive phenotyping of both investigated cohorts, we examined the interrelation between MetS, cognitive function, and brain structure in a mediation analysis. Finally, to uncover factors associated with brain region-specific MetS effects, we mapped local cellular as well as network topological attributes to observed MetS-associated cortical abnormalities. With this work, we aimed to advance the understanding of the fundamental principles underlying the neurobiology of MetS.

## Results

### Sample characteristics

Application of exclusion criteria and quality assessment ruled out 2188 UKB subjects and 30 HCHS subjects resulting in a final analysis sample of 40,087 individuals. For a flowchart providing details on the sample selection procedure please refer to *Appendix 1—figure 1*. Descriptive statistics are listed in *Table 1*. To sensitivity analyze our results, as well as to facilitate the comparison with previous reports which primarily rely on a case-control design, we supplemented group statistics comparing individuals with clinically defined MetS and matched controls, where applicable. Corresponding group analysis results are described in more detail in *appendix 2*.

### Partial least squares correlation analysis

We investigated the relationship between brain morphological and clinical measures of MetS (abdominal obesity, arterial hypertension, dyslipidemia, insulin resistance) in a PLS considering all individuals from both studies (n=40,087) (*Figure 1*). By this, we aimed to detect the continuous effect of any MetS component independent from a formal binary classification of MetS (present/not present). A correlation matrix relating all considered MetS component measures is displayed in *Appendix 1—figure 2*. Before conducting the PLS, brain morphological and clinical data were deconfounded for age, sex, education, and cohort effects.

PLS identified eight significant latent variables which represent clinical-anatomical dimensions relating MetS components to brain morphology (*Appendix 1—table 1*). The first latent variable explained 71.20% of the shared variance and was thus further investigated (*Figure 2a*). Specifically, the first latent variable corresponded with a covariance profile of lower severity of MetS (*Figure 2c*; loadings [95% confidence interval]; waist circumference: –0.230 [–0.239, –0.221], hip circumference: –0.187 [–0.195, –0.178], waist-hip ratio: –0.167 [–0.176, –0.158], body mass index: –0.234 [–0.243, –0.226], systolic blood pressure: –0.089 [–0.098, –0.080], diastolic blood pressure: –0.116 [–0.125, –0.107], high-density lipoprotein: 0.099 [0.090, 0.108], low-density lipoprotein: –0.013 [–0.022, –0.004], total cholesterol: 0.003 [–0.006, 0.012], triglycerides: –0.102 [–0.111, –0.092], HbA1c: –0.064 [–0.073, –0.54], glucose: –0.049 [–0.058, –0.039]). Notably, the obesity-related measures showed the strongest contribution to the covariance profile as indicated by the highest loading to the latent variable. Age (<0.001 [–0.009, 0.009]), sex (<0.001 [–0.009, 0.009]), education (<0.001 [–0.009, 0.009]), and cohort (<–0.001 [–0.008, 0.007]) did not significantly contribute to the latent variable, which is compatible with sufficient effects of deconfounding. Details on the second latent variable which explained 22.33% of shared variance are provided in *Figure 2—figure supplement 1*. In brief, it predominantly related lower HbA1c and blood glucose to higher thickness and volume in lateral frontal, posterior temporal, parietal, and occipital regions and vice versa.

Bootstrap ratios (= $\frac{singular\ vector\ weight}{bootstrap-estimated\ standard\ error}$) were computed to identify brain regions with a significant contribution to the covariance profile (see Methods). Cortical thickness in orbitofrontal, lateral prefrontal, insular, anterior cingulate, and temporal areas as well as volumes of all investigated subcortical regions contributed positively to the covariance profile as indicated by a positive bootstrap ratio (*Figure 2d*). Thus, a higher cortical thickness and subcortical volume in these areas corresponded with less obesity, hypertension, dyslipidemia, and insulin resistance and vice versa, i.e., lower cortical thickness and subcortical volumes with increased severity of MetS. A negative bootstrap ratio was found in superior frontal, parietal, and occipital regions indicating that a higher cortical thickness in these regions corresponded with more severe MetS. This overall pattern was confirmed via conventional, vertex-wise group comparisons of cortical thickness measurements based on the binary classification of individuals with MetS and matched controls (*Appendix 2—figure 4*) as well as subsample analyses considering the UKB and HCHS participants independently (*Figure 2—figure supplements*

**Table 1.** Descriptive statistics UKB and HCHS.

| Metric | Stat* |
|---|---|
| Age (years) | 63.55±7.59 (40087) |
| Sex (% female) | 46.47 (40087) |
| Education (ISCED) | 2.62±0.73 (39944) |
| | |
| Metabolic syndrome components | |
| Waist circumference (cm) | 88.47±12.71 (38800) |
| Hip circumference (cm) | 100.90±8.79 (38801) |
| Waist-hip ratio | 0.88±0.09 (38800) |
| Body mass index | 26.47±4.37 (38701) |
| $RR_{systolic}$ (mmHg) | 138.30±18.57 (31234) |
| $RR_{diastolic}$ (mmHg) | 78.88±10.09 (31238) |
| Antihypertensive therapy (%) | 6.96 (39976) |
| HDL (mg/dL) | 61.76±23.69 (34468) |
| LDL (mg/dL) | 137.38±36.29 (37456) |
| Cholesterol (mg/dL) | 211.29±56.42 (37531) |
| Triglycerides (mg/dL) | 148.90±83.84 (37510) |
| Lipid lowering therapy (%) | 14.44 (39976) |
| HbA1c (%) | 5.37±0.48 (37284) |
| Blood glucose (mg/dL) | 90.29±17.58 (34432) |
| Antidiabetic therapy (%) | 0.45 (39976) |
| | |
| Imaging | |
| Mean cortical thickness (mm) | 2.40±0.09 (40087) |
| | |
| Cognitive variables of the UK Biobank | |
| Fluid Intelligence | 6.63±2.06 (36510) |
| Matrix Pattern Completion | 7.99±2.13 (25771) |
| Numeric Memory Test | 6.69±1.52 (26780) |
| Paired Associate Learning | 6.92±2.63 (26048) |
| Prospective Memory | 1.07±0.39 (37192) |
| Reaction Time (sec) | 594.16±109.08 (37015) |
| Symbol Digit Substitution | 18.96±5.25 (25810) |
| Tower Rearranging Test | 9.91±3.23 (25555) |
| Trail Making Test A (sec) | 223.03±86.51 (26048) |
| Trail Making Test B (sec) | 550.01±270.09 (26048) |
| | |
| Cognitive variables of the Hamburg City Health Study | |
| Animal Naming Test | 24.78±6.92 (2416) |
| Clock Drawing Test | 6.43±1.12 (2479) |

*Table 1 continued on next page*

*Table 1 continued*

| Metric | Stat* |
|---|---|
| Trail Making Test A (sec) | 40.09±14.33 (2290) |
| Trail Making Test B (sec) | 90.05±37.30 (2264) |
| Multiple-Choice Vocabulary Intelligence Test | 31.27±3.58 (2026) |
| Word List Recall | 7.75±1.84 (2342) |

*Presented as mean ± SD (N).

**2 and 3**). The correlation matrix of all spatial effect maps investigated in this study (bootstrap ratio and Schaefer 400-parcellated t-statistic from group comparisons) is visualized in **Figure 2—figure supplement 4**. All derived effect size maps were significantly correlated ($r_{sp}$ =0.67−0.99, $p_{FDR}$ < 0.05) (**Schaefer et al., 2018**).

Subject-specific imaging and clinical scores for the first latent variable were computed. These scores indicate to which degree an individual expresses the corresponding covariance profiles. By definition, the scores are correlated ($r_{sp}$ = 0.201, p<0.005, **Figure 2b**) indicating that individuals exhibiting the clinical covariance profile (severity of MetS components) also express the brain morphological pattern. This relationship was robust across a 10-fold cross-validation (avg. $r_{sp}$ = 0.19, **Appendix 1—table 2**).

These results were consistent in separate PLS analyses for both the UKB and HCHS samples, as displayed in **Figure 2—figure supplements 2 and 3**. In these subset-specific analyses, cognitive test performances significantly contributed to the first latent variable when included in the PLS. Consequently, the first latent variable associated more severe MetS with both brain morphological abnormalities and poorer cognitive performance.

## Mediation analysis of cognitive outcomes

To gain a better understanding of the link between MetS, brain morphology, and cognitive function, we performed a mediation analysis on cognitive test results and subject-specific PLS scores. Therefore, we investigated whether the imaging PLS score (representing MetS-related brain structural abnormalities) acts as a mediator in the relationship between the clinical PLS score (representing MetS severity) and cognitive test performances. Importantly, scores of the main PLS analysis, which did not include cognitive measures, were considered. The corresponding path plots are shown in **Figure 3**. The imaging score was found to fully mediate the relationship of the clinical score and results of the Trail Making Test B (ab = –0.011, $P_{FDR}$ <0.001; c'=–0.012, $P_{FDR}$ = 0.072; c=–0.023, $P_{FDR}$ <0.001), Fluid Intelligence Test (ab = 0.017, $P_{FDR}$ <0.001; c'=0.011, $P_{FDR}$ = 0.072; c=0.028, $P_{FDR}$ <0.001) as well as Matrix Pattern Completion Test (ab = 0.015, $P_{FDR}$ <0.001; c'=0.010, $P_{FDR}$ = 0.172; c=0.025, $P_{FDR}$ <0.001). Further, the imaging score partially mediated the relationship of the clinical score and results of the Symbol Digit Substitution Test (ab = 0.010, $P_{FDR}$ <0.001; c'=0.036, $P_{FDR}$ <0.001; c=0.046, $P_{FDR}$ <0.001), Numeric Memory Test (ab = 0.014, $P_{FDR}$ <0.001; c'=0.044, $P_{FDR}$ <0.001; c=0.058, $P_{FDR}$ <0.001) and Paired Associate Learning Test (ab = 0.015, $P_{FDR}$ <0.001; c'=0.044, $P_{FDR}$ <0.001; c=0.059, $P_{FDR}$ <0.001). For the remaining cognitive tests, no significant mediation was found.

## Contextualization of MetS-associated brain morphological abnormalities

We investigated whether the pattern of MetS effects on cortical structure is linked to the regional density of specific cell populations and global brain network topology in a surface-based contextualization analysis (see Methods).

Therefore, we first used a virtual histology approach to relate the bootstrap ratio from PLS to the differential expression of cell-type specific genes based on microarray data from the Allen Human Brain Atlas (**Hawrylycz et al., 2012**). The results are illustrated in **Figure 4**. The bootstrap ratio was significantly positively correlated with the density of endothelial cells ($Z_{r_{sp}}$ = 0.190, $p_{FDR}$ = 0.016), microglia ($Z_{r_{sp}}$ = 0.271, $p_{FDR}$ = 0.016), excitatory neurons type 8 ($Z_{r_{sp}}$ = 0.165, $p_{FDR}$ = 0.016), inhibitory neurons type 1 ($Z_{r_{sp}}$ = 0.363, $p_{FDR}$ = 0.036) and excitatory neurons type 6 ($Z_{r_{sp}}$ = 0.146, $p_{FDR}$ = 0.034) indicating that MetS-related brain morphological abnormalities are strongest in regions of the

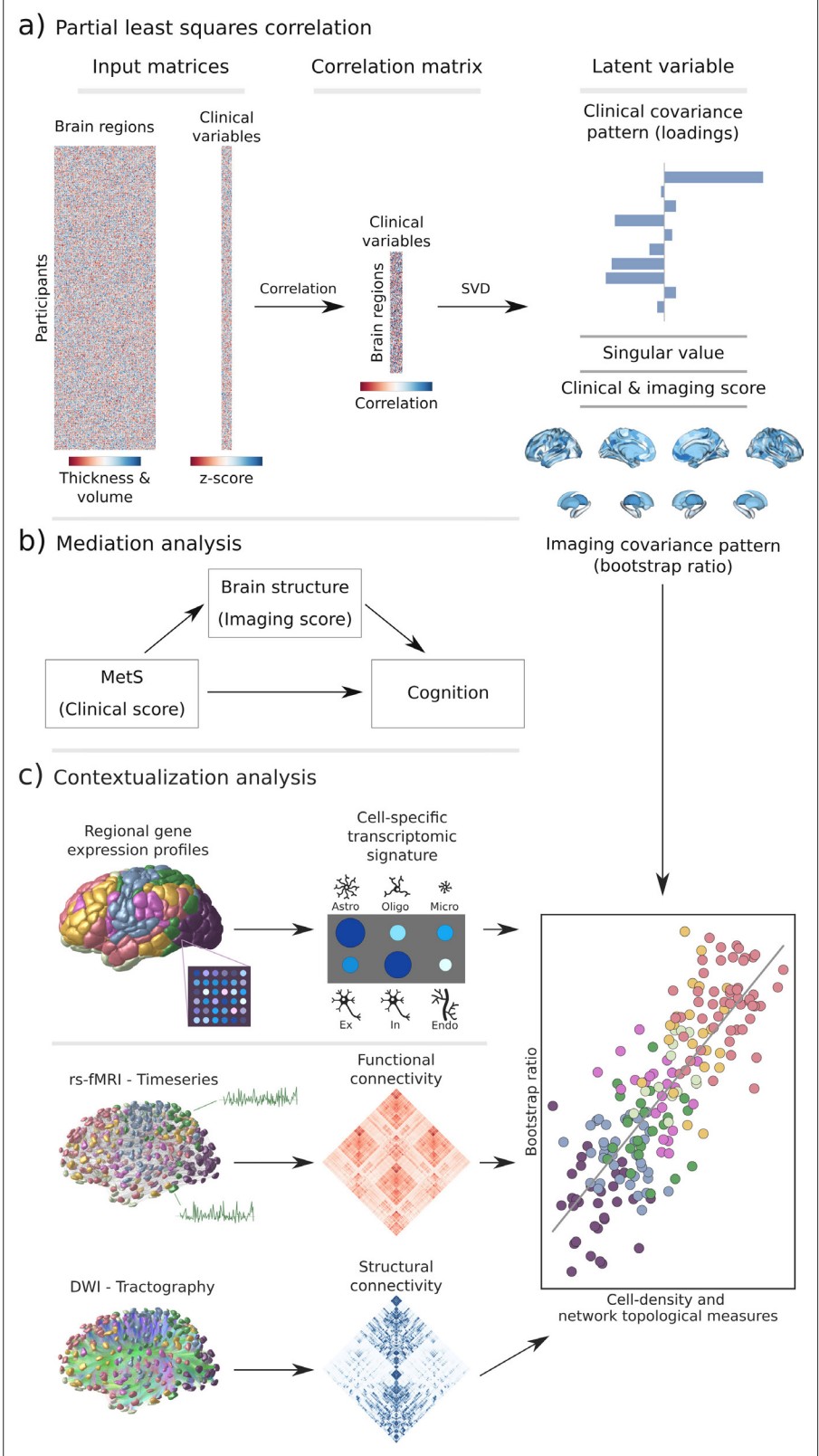

**Figure 1.** Methodology. (**a**) Illustration of the partial least squares correlation analysis. Starting from two input matrices containing per-subject information of regional morphological measures as well as clinical data (demographic and metabolic syndrome (MetS)-related risk factors) a correlation matrix is computed. This matrix is subsequently subjected to singular value decomposition resulting in a set of mutually orthogonal latent

*Figure 1 continued on next page*

*Figure 1 continued*

variables. Latent variables each consist of a left singular vector (here, clinical covariance profile), singular value, and right singular vector (here, imaging covariance profile). In addition, subject-specific clinical and imaging scores are computed. (**b**) The interplay between MetS, brain structure, and cognition was investigated in a post-hoc mediation analysis. We tested whether the relationship between the clinical score, representing MetS severity, and different cognitive test performances was statistically mediated by the imaging score. (**c**) Contextualization analysis. Upper row: based on microarray gene expression data, the densities of different cell populations across the cortex were quantified. Middle and lower row: based on functional and structural group-consensus connectomes based on data from the Human Connectome Project, metrics of functional and structural brain network topology were derived. Cell density as well as connectomic measures were related to the bootstrap ratio via spatial correlations. Modified from *Petersen et al., 2022b*; *Zeighami et al., 2019*. Abbreviations: Astro – astrocytes; DWI – diffusion-weighted magnetic resonance imaging; Endo – endothelial cells; Ex – excitatory neuron populations (Ex1-8); In – inhibitory neuron populations (In1-8); Micro – microglia; Oligo – oligodendrocytes; rs-fMRI – resting-state functional magnetic resonance imaging; SVD – singular value decomposition.

highest density of these cell types. No significant associations were found regarding the remaining excitatory neuron types (Ex1-Ex5, Ex7), inhibitory neurons (In2-In8), astrocytes, and oligodendrocytes (*Appendix 1—table 3*). Virtual histology analysis results for bootstrap ratios corresponding with latent variables 2 and 3 are shown in *Figure 4—figure supplement 1*. As a sensitivity analysis, we contextualized the t-statistic map derived from group statistics. The results remained stable except for excitatory neurons type 6 ($Z_{r_{sp}} = 0.145$, $p_{FDR} = 0.123$) and inhibitory neurons type 1 ($Z_{r_{sp}} = 0.432$, $p_{FDR} = 0.108$), which no longer showed a significant association (*Figure 4—figure supplement 2*, *Appendix 1—table 4*).

Second, we associated the bootstrap ratio with three pre-selected measures of brain network topology derived from group consensus functional and structural connectomes of the Human Connectome Project (HCP) (*Figure 5*): weighted degree centrality (marking brain network hubs), neighborhood abnormality, and macroscale functional connectivity gradients (*Petersen et al., 2022b*). The bootstrap ratio showed a medium positive correlation with the functional neighborhood abnormality ($r_{sp} = 0.464$, $p_{spin} < 0.001$, $p_{smash} < 0.001$, $p_{rewire} < 0.001$) and a strong positive correlation with the structural neighborhood abnormality ($r_{sp} = 0.764$, $p_{spin} = {<}0.001$, $p_{smash} < 0.001$, $p_{rewire} < 0.001$) indicating functional and structural interconnectedness of areas exhibiting similar MetS effects. These results remained significant when the t-statistic map was contextualized instead of the bootstrap ratio as well as when neighborhood abnormality measures were derived from consensus connectomes of the HCHS instead of the HCP (*Figure 5—figure supplements 1 and 2*). We found no significant associations for the remaining indices of network topology, i.e., functional degree centrality ($r_{sp} = 0.163$, $p_{spin} = 0.365$, $p_{smash} = 0.406$, $p_{rewire} = 0.870$), structural degree centrality ($r_{sp} = 0.029$, $p_{spin} = 0.423$, $p_{smash} = 0.814$, $p_{rewire} = 0.103$) as well as functional cortical gradient 1 ($r_{sp} = 0.152$, $p_{spin} = 0.313$, $p_{smash} = 0.406$, $p_{rewire} = 0.030$) and gradient 2 ($r_{sp} = -0.177$, $p_{spin} = 0.313$, $p_{smash} = 0.406$, $p_{rewire} < 0.001$).

## Discussion

We investigated the impact of MetS on brain morphology and cognitive function in a large sample of individuals from two population-based neuroimaging studies. We report three main findings: (1) multivariate, data-driven statistics revealed a latent variable relating MetS and brain health: participants were distributed along a clinical-anatomical dimension of interindividual variability, linking more severe MetS to widespread brain morphological abnormalities. Negative MetS-related brain morphological abnormalities were strongest in orbitofrontal, lateral prefrontal, insular, cingulate, and temporal cortices as well as subcortical areas. Positive MetS-related brain morphological abnormalities were strongest in superior frontal, parietal, and occipital regions. (2) The severity of MetS was associated with executive function and processing speed, memory, and reasoning test performances, and was found to be statistically mediated by MetS-related brain morphological abnormalities. (3) The pattern of MetS-related brain morphological abnormalities appeared to be linked to regional cell composition as well as functional and structural connectivity. These findings were robust across sensitivity analyses. In sum, our study provides an in-depth examination of the intricate relationship between MetS, brain morphology, and cognition. A graphical abstract summarizing the results is included as *Figure 6*.

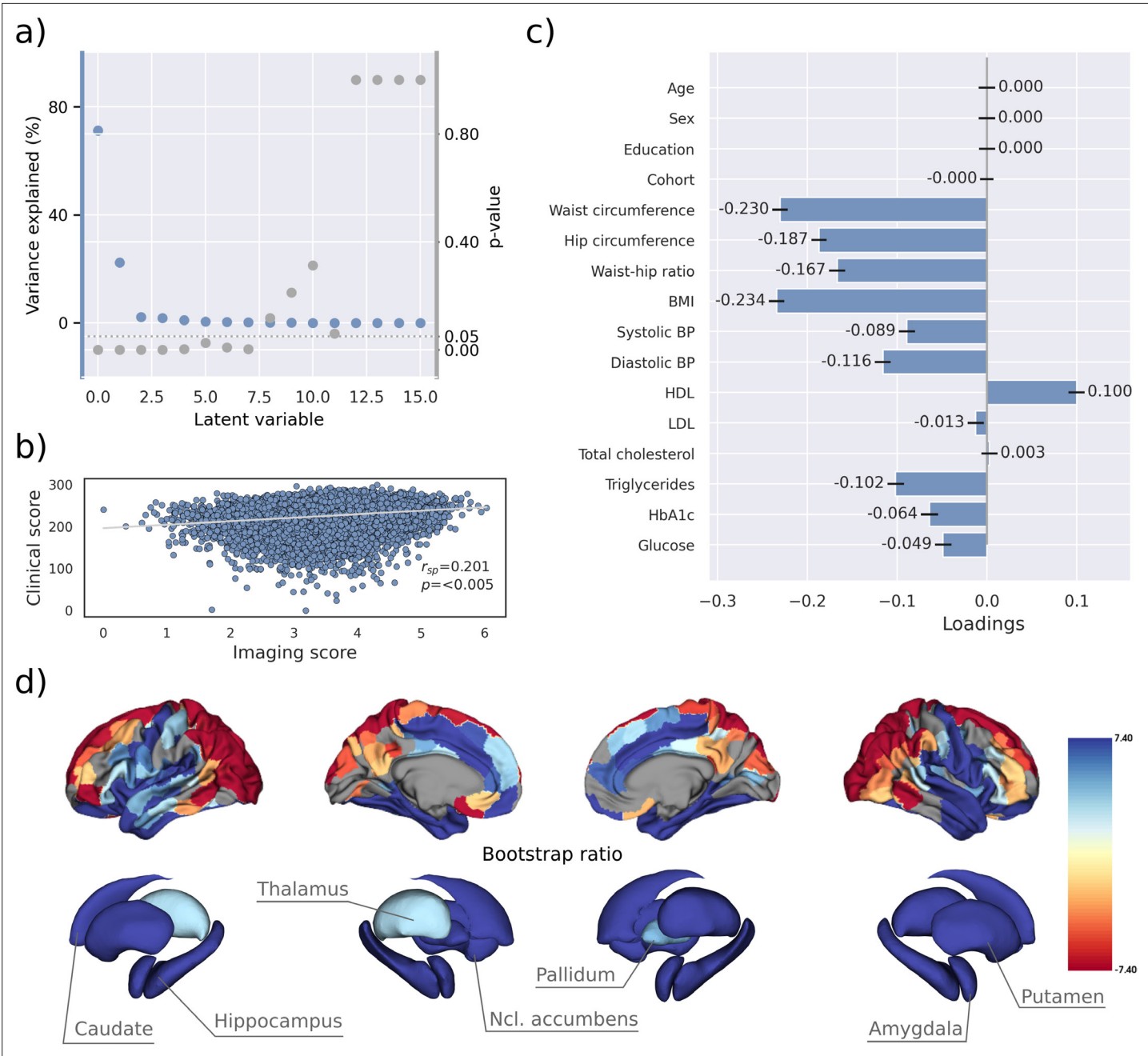

**Figure 2.** Partial least squares correlation analysis (PLS). (**a**) Explained variance and p-values of latent variables. (**b**) Scatter plot relating subject-specific clinical and imaging PLS scores. Higher scores indicate higher adherence to the respective covariance profile. (**c**) Clinical covariance profile. 95% confidence intervals were calculated via bootstrap resampling. Note that confound removal for age, sex, education, and cohort was performed prior to the PLS. (**d**) Imaging covariance profile represented by bootstrap ratio. A high positive or negative bootstrap ratio indicates high contribution of a brain region to the overall covariance profile. Regions with a significant bootstrap ratio (>1.96 or <–1.96) are highlighted by colors. Abbreviations: BMI – Body mass index, HDL – high-density lipoprotein, LDL – low-density lipoprotein, $r_{sp}$ - Spearman correlation coefficient.

The online version of this article includes the following figure supplement(s) for figure 2:

**Figure supplement 1.** Partial least squares correlation analysis – Latent variable 2.

**Figure supplement 2.** Partial least squares correlation analysis – UK Biobank (including cognitive test results).

**Figure supplement 3.** Partial least squares correlation analysis – Hamburg City Health Study (HCHS) (including cognitive test results).

**Figure supplement 4.** Spatial correlation of effect size maps.

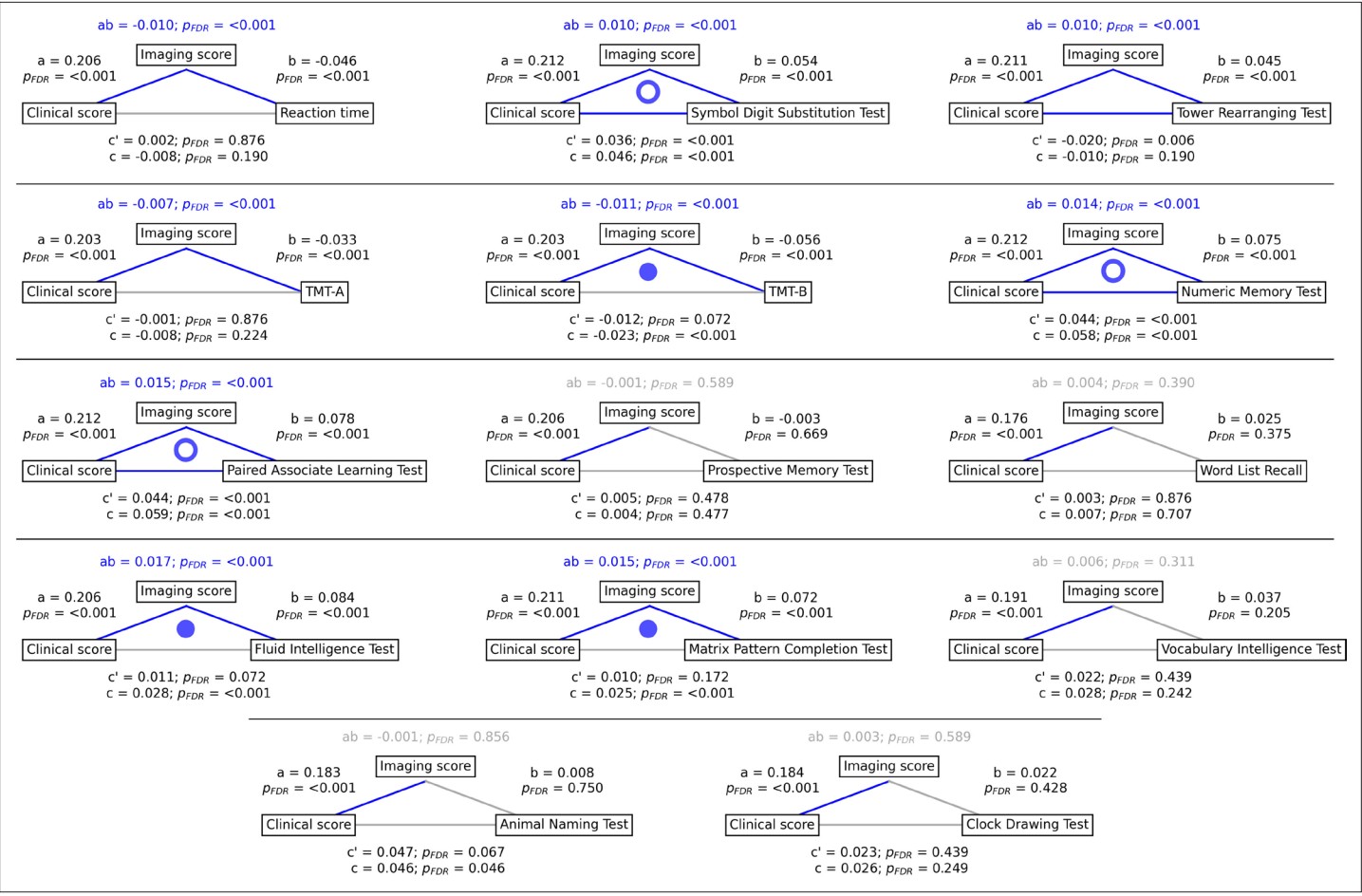

**Figure 3.** Mediation analysis results. Mediation effects of subject-specific imaging PLS scores on the relationship between metabolic syndrome (MetS) represented by the clinical PLS score and cognitive test performances. Path plots display standardized effects and p-values: (**a**) clinical score to imaging score, (**b**) imaging score to cognitive score, (ab) indirect effect (**c'**) direct effect, and (**c**) total effect. Significant paths are highlighted in blue; non-significant in light gray. If the indirect effect ab was significant, the text for ab is highlighted in blue. A blue dot in the path plot indicates if a relationship is significantly mediated, i.e., the indirect effect ab was significant and the direct effect c' was reduced or non-significant compared to the total effect c. An empty dot indicates a partial mediation, and a full dot indicates a full mediation. Abbreviations: $p_{FDR}$ - false discovery rate-corrected p-values; PLS – partial least squares correlation; TMT-A – Trail Making Test A; TMT-B – Trail Making Test B.

## PLS reveals a latent clinical-anatomical dimension relating MetS and brain health

MetS adversely impacts brain health through complex, interacting effects on the cerebral vasculature and parenchyma as shown by histopathological and imaging studies (*Borshchev et al., 2019*). The pathophysiology of MetS involves atherosclerosis, which affects blood supply and triggers inflammation (*Libby et al., 2002*; *Birdsill et al., 2013*); endothelial dysfunction reducing cerebral vasoreactivity *Lind, 2008*; breakdown of the blood-brain barrier inciting an inflammatory response *Hussain et al., 2021*; oxidative stress causing neuronal and mitochondrial dysfunction *Mullins et al., 2020*; and small vessel injury leading to various pathologies including white matter damage, microinfarcts, and cerebral microbleeds (*Frey et al., 2019*).

To address these interacting effects, we harnessed multivariate, data-driven statistics in the form of a PLS in two large-scale population-based studies to probe for covariance profiles relating the full range of MetS components (such as obesity or arterial hypertension) to regional brain morphological information in a single analysis. PLS identified eight significant latent variables with the first variable explaining the majority (71.20%) of shared variance within the imaging and clinical data (*Figure 2a*). This finding indicates a relatively uniform connection between MetS and brain morphology, implying

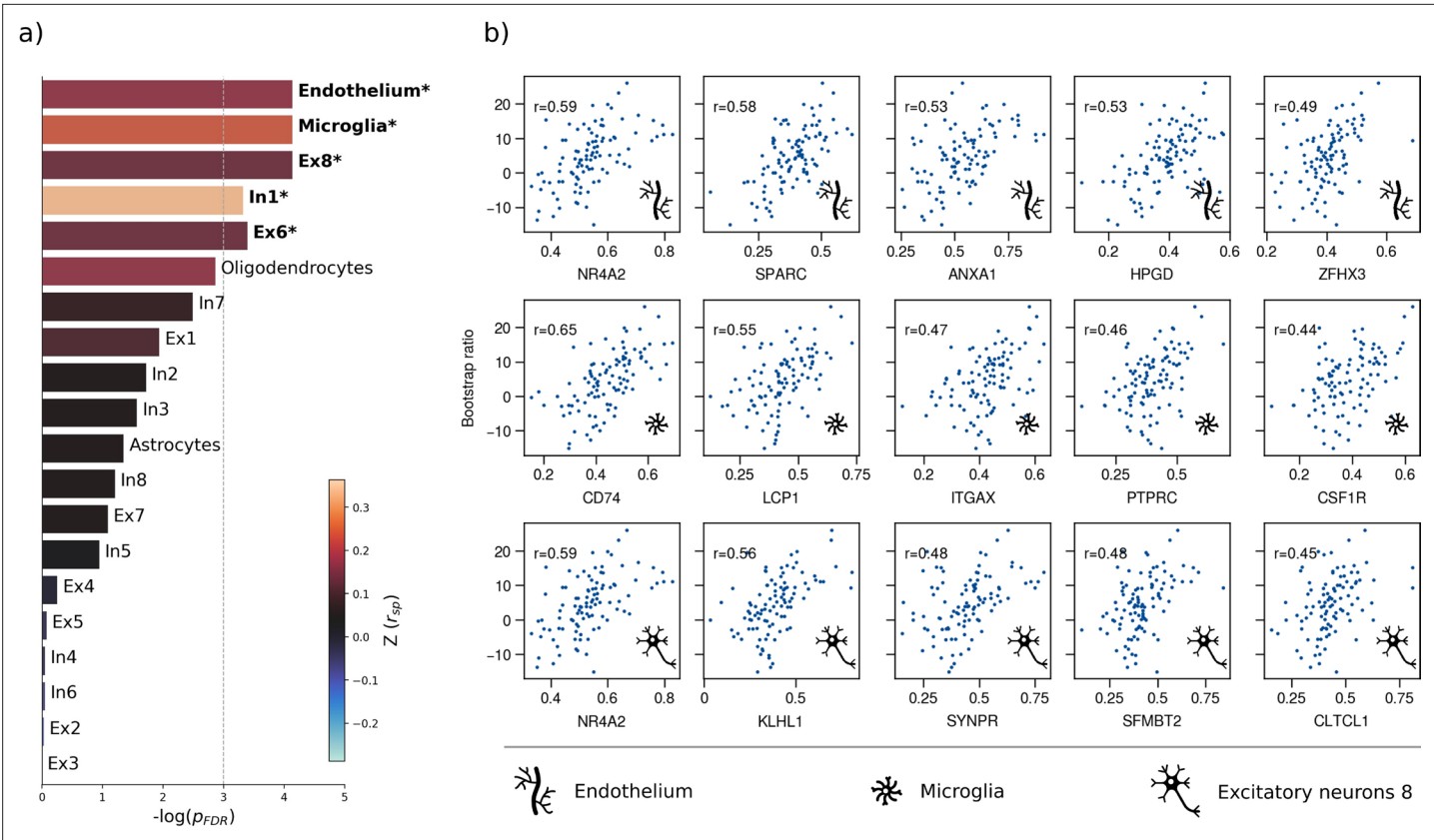

**Figure 4.** Virtual histology analysis. The regional correspondence between metabolic syndrome (MetS) effects (bootstrap ratio) and cell type-specific gene expression profiles was examined via an ensemble-based gene category enrichment analysis. (**a**) Barplot displaying spatial correlation results. The bar height displays the significance level. Colors encode the aggregate z-transformed Spearman correlation coefficient relating the Schaefer100-parcellated bootstrap ratio and respective cell population densities. Asterisks indicate statistical significance. The significance threshold of $p_{FDR} < 0.05$ is highlighted by a vertical dashed line. (**b**) Scatter plots illustrating spatial correlations between MetS effects and exemplary cortical gene expression profiles per cell population significantly associated across analyses – i.e., endothelium, microglia, and excitatory neurons type 8. Top 5 genes most strongly correlating with the bootstrap ratio map were visualized for each of these cell populations. Icons in the bottom right of each scatter plot indicate the corresponding cell type. A legend explaining the icons is provided at the bottom. First row: endothelium; second row: microglia; third row: excitatory neurons type 8. Virtual histology analysis results for the bootstrap ratios of latent variables 2 and 3 are shown in *Figure 4—figure supplement 1*. A corresponding plot illustrating the contextualization of the t-statistic derived from group statistics is shown in *Figure 4—figure supplement 2*. Abbreviations: $-log\left(p_{FDR}\right)$ – negative logarithm of the false discovery rate-corrected p-value derived from spatial lag models (*Dukart et al., 2021*; *Burt et al., 2018*); $r$ – Spearman correlation coeffient. $Z\left(r_{sp}\right)$ – aggregate z-transformed Spearman correlation coefficient.

The online version of this article includes the following figure supplement(s) for figure 4:

**Figure supplement 1.** Virtual histology analysis of latent variables 2 and 3.

**Figure supplement 2.** Sensitivity virtual histology analysis based on t-statistic map from group comparison.

that the associative effects of various MetS components on brain structure are comparatively similar, despite the distinct pathomechanisms each component entails.

PLS revealed that all MetS components were contributing to this latent signature. However, waist circumference, hip circumference, waist-hip ratio, and body mass index consistently contributed higher than the remaining variables across conducted analyses which highlights obesity as the strongest driver of MetS-related brain morphological abnormalities.

We interpret these findings as evidence that MetS-associated conditions jointly contribute to the harmful effects on brain structure rather than affecting it in a strictly individual manner. This notion is supported by previous work in the UKB demonstrating overlapping effects of individual risk factors on brain morphology (*Cox et al., 2019*). Specifically, the first latent variable related to increased severity of obesity, dyslipidemia, arterial hypertension, and insulin resistance with lower thickness in orbitofrontal, lateral prefrontal, insular, cingulate, and temporal cortices as well as lower volume

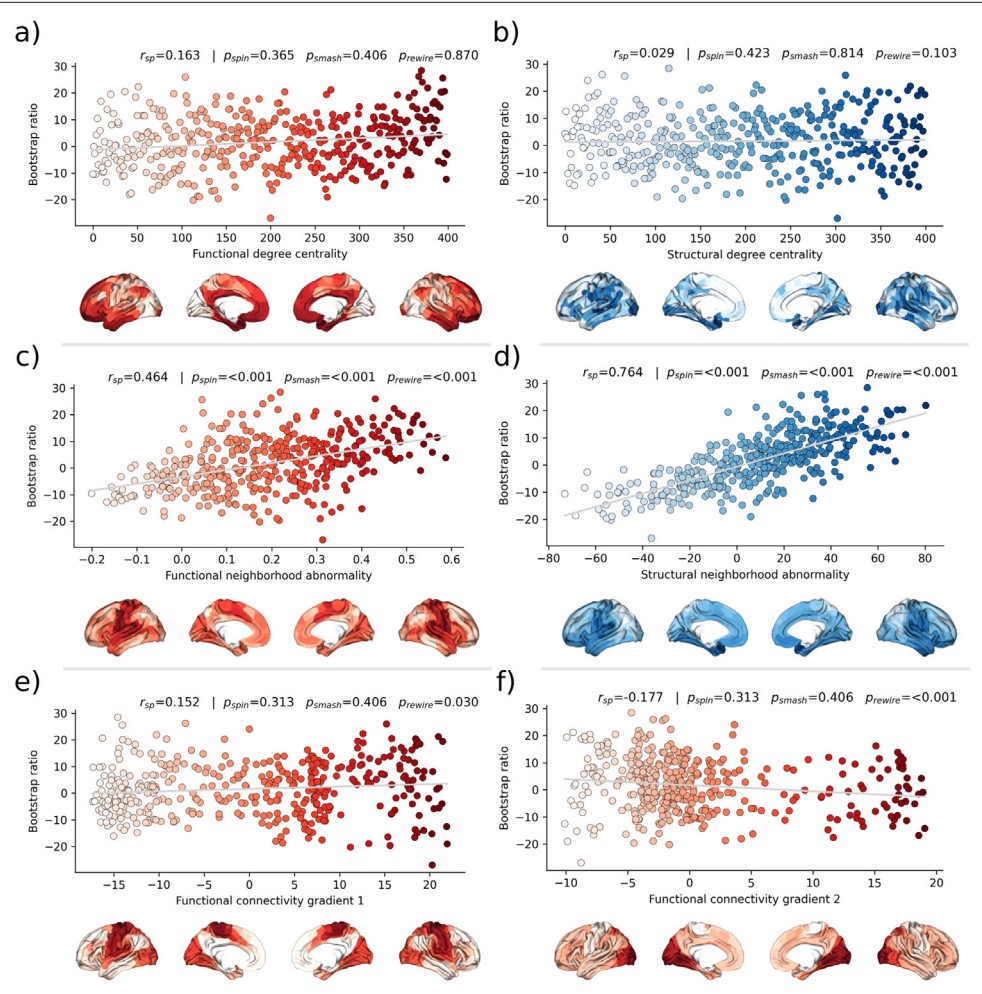

**Figure 5.** Brain network contextualization. Spatial correlation results derived from relating Schaefer 400×7-parcellated maps of metabolic syndrome (MetS) effects (bootstrap ratio) to network topological indices (red: functional connectivity, blue: structural connectivity). Scatter plots that illustrate the spatial relationship are supplemented by surface plots for anatomical localization. The color coding of cortical regions and associated dots corresponds. (**a and b**) Functional and structural degree centrality rank. (**c and d**) Functional and structural neighborhood abnormality. (**e and f**) Intrinsic functional network hierarchy represented by functional connectivity gradients 1 and 2. Complementary results concerning t-statistic maps derived from group comparisons between MetS subjects and controls are presented in *Figure 5—figure supplement 1*. Corresponding results after reperforming the analysis with HCHS-derived group-consensus connectomes are presented in *Figure 5—figure supplement 2*. Abbreviations: HCHS – Hamburg City Health Study; $p_{rewire}$ - p-value derived from network rewiring (*Maslov et al., 2004*); $p_{smash}$ - p-value derived from brainSMASH surrogates (*Burt et al., 2020*); $p_{spin}$ - p-value derived from spin permutation results (*Alexander-Bloch et al., 2018*); $r_{sp}$ - Spearman correlation coefficient.

The online version of this article includes the following figure supplement(s) for figure 5:

**Figure supplement 1.** Sensitivity network contextualization analysis based on t-statistic map derived from group comparison.

**Figure supplement 2.** Sensitivity network contextualization analysis based on group-consensus connectomes from the Hamburg City Health Study.

across subcortical regions (*Figure 2c and d*). This profile was consistent in separate PLS analyses of UKB and HCHS participants as well as group comparisons (*Figure 2—figure supplement 4*). Previous research aligns with our detection of a MetS-associated frontotemporal morphometric abnormality pattern (*Beyer et al., 2019*; *McIntosh et al., 2017*; *Kotkowski et al., 2019*). As a speculative causative pathway, human and animal studies have related the orbitofrontal, insular, and anterior cingulate cortex to food-related reward processing, taste, and impulse regulation (*Tuulari et al., 2015*; *Rolls,*

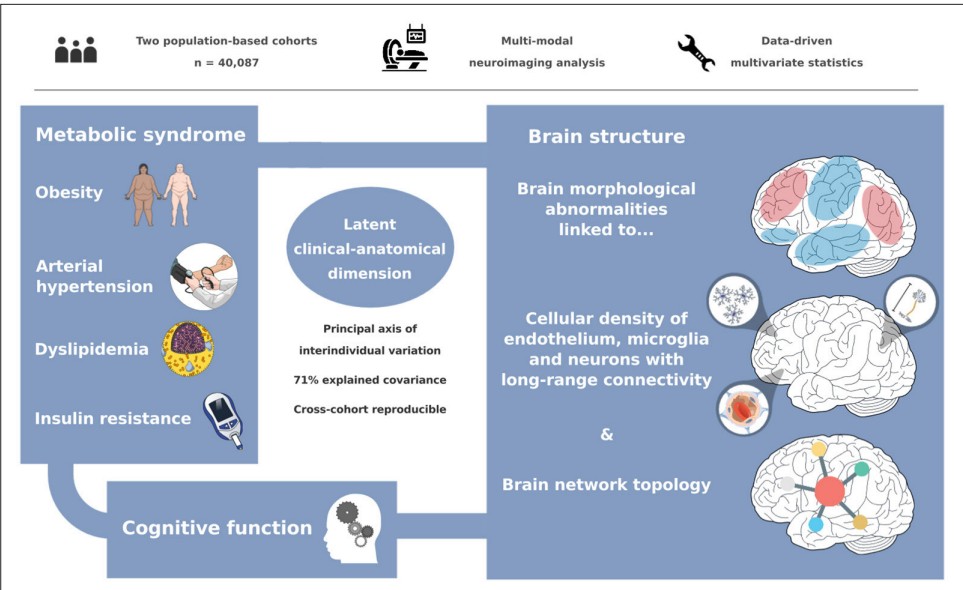

**Figure 6.** Graphical abstract.

*2016*). Conceivably, structural alterations of these brain regions are linked to brain functions and behaviors that exacerbate the risk profile leading to MetS (*Rolls, 2023*; *Price et al., 2019*). We also noted a positive MetS-cortical thickness association in superior frontal, parietal, and occipital lobes, a less intuitive finding that has been previously reported (*Krishnadas et al., 2013*; *Leritz et al., 2011*). Although speculative, the positive effects might be due to MetS compensating cholesterol disruptions associated with neurodegenerative processes (*Qin et al., 2021*).

The second latent variable accounted for 22.33% of the shared variance and linked higher markers of insulin resistance and lower dyslipidemia to lower thickness and volume in lateral frontal, posterior temporal, parietal, and occipital regions. The distinct covariance profile of this latent variable, compared to the first, likely indicates a separate pathomechanistic connection between MetS components and brain morphology. Given that HbA1c and blood glucose were the most significant contributors to this variable, insulin resistance might drive the observed clinical-anatomical relationship.

## Brain morphological abnormalities mediate the relationship between MetS and cognitive deficits

Cognitive performance has been consistently linked to cardiometabolic risk factors in health and disease (*Genon et al., 2022*). Yet, the pathomechanistic correlates of this relationship remain to be understood. Our mediation analysis revealed that increased MetS severity correlates with worse performance in executive function and processing speed (Symbol Digit Substitution Test, Trail Making Test B), memory (Numeric Memory Test, Paired Associate Learning Test), and reasoning (Fluid intelligence, Matrix Pattern Completion Test), with brain morphological abnormalities statistically mediating these relationships. Additionally, group comparisons indicated poorer cognitive performance in MetS subjects (*Appendix 2—tables 1 and 2*) and including cognitive outcomes in the PLS as clinical variables revealed a significant contribution to the first latent variable (*Figure 2—figure supplements 2 and 3*). These results suggest that MetS is significantly associated with cognitive deficits across various domains, and brain morphological abnormalities are a crucial pathomechanistic link in this relationship. In support of this, previous studies have shown that brain structure mediates the relationship between MetS and cognitive performance in a pediatric sample and elderly patients with vascular cognitive impairment (*Laurent et al., 2020*; *Seo et al., 2010*; *Kim et al., 2014*). The detected latent variable might represent a continuous disease spectrum spanning from minor cognitive deficits due to a cardiometabolic risk profile to severe cognitive deficits due to dementia. In support of this hypothesis, the determined brain morphological abnormality pattern is consistent with the atrophy pattern

found in vascular mild cognitive impairment, vascular dementia, and Alzheimer's dementia (*Seo et al., 2010*; *Kim et al., 2014*; *Morys et al., 2023*).

Collectively, these findings highlight the role of MetS in cognitive impairment and underscore the potential impact of therapies targeting cardiometabolic risk factors. Although the definitive role of such therapies in preventing cognitive decline is not yet fully established, emerging evidence suggests that these interventions can mitigate the adverse cognitive effects of MetS (*Veronese et al., 2017*; *Lennon et al., 2023*; *Gelber et al., 2013*). As our results highlight obesity as a key factor in the observed clinical-anatomical relationship, we think that future studies should further investigate weight-reducing interventions to examine their effects on cognitive outcomes. Advanced neuro-imaging techniques promise to refine these therapeutic approaches by enabling to identify MetS patients at risk of cognitive decline that would benefit the most from targeted interventions for cognitive health protection.

## MetS-related brain morphological abnormalities link to cellular tissue composition and network topology

To better understand the emergence of the spatial pattern of MetS-related brain morphological abnormalities, we conducted two contextualization analyses leveraging reference datasets of local gene expression data as well as properties of brain network topology.

Using a virtual histology approach based on regional gene expression data, we investigated MetS effects in relation to cell population densities (*Figure 4*). As the main finding, we report that higher MetS-related brain morphological abnormalities coincide with a higher regional density of endothelial cells. This aligns with the known role of endothelial dysfunction in MetS compromising tissues via chronic vascular inflammation, increased thrombosis risk, and hypoperfusion due to altered vasoreactivity and vascular remodeling (*Lind, 2008*). As endothelial density also indicates the degree of general tissue vascularization, well-vascularized regions are also likely more exposed to cardiometabolic risk factor effects in general (*Libby et al., 2002*). Our results furthermore indicate that microglial density determines a brain region's susceptibility to MetS effects. Microglia are resident macrophages of the central nervous system that sustain neuronal integrity by maintaining a healthy microenvironment. Animal studies have linked microglial activation mediated by blood-brain barrier leakage and systemic inflammation to cardiometabolic risk (*Denes et al., 2012*; *Tucsek et al., 2014*). Activated microglia can harm the brain structure by releasing reactive oxygen species, proinflammatory cytokines, and proteinases (*Dheen et al., 2007*). Lastly, we found an association with the density of excitatory neurons of subtype 8. These neurons reside in cortical layer 6 and their axons mainly entertain long-range cortico-cortical and cortico-thalamic connections (*Lake et al., 2016*; *Thomson, 2010*). Consequently, layer 6 neurons might be particularly susceptible to MetS effects due to their exposition to MetS-related white matter disease (*Petersen et al., 2022a*; *Petersen et al., 2020*). Taken together, the virtual histology analysis indicates that MetS-related brain morphological abnormalities are associated with local cellular fingerprints. Our findings emphasize the involvement of endothelial cells and microglia in brain structural abnormalities due to cardiometabolic risk, marking them as potential targets for therapies aimed at mitigating MetS effects on brain health.

For the second approach, we contextualized MetS-related brain morphological abnormalities using principal topological properties of functional and structural brain networks. We found that regional MetS effects and those of functionally and structurally connected neighbors were correlated (*Figure 5c and d*) – i.e., areas with similar MetS effects tended to be disproportionately interconnected. Put differently, MetS effects coincided within functional and structural brain networks. Therefore, our findings can be interpreted as evidence that a region's functional and structural network embedding – i.e., its individual profile of functional interactions as well as white matter fiber tract connections – are associated with its susceptibility to morphological MetS effects. Multiple mechanisms might explain how connectivity might be associated with MetS-related morphological alterations. For example, microvascular pathology might impair white matter fiber tracts leading to joint degeneration in interconnected cortical brain areas: that is, the occurrence of shared MetS effects within functionally and structurally connected neighborhoods is explained by their shared (dis-)connectivity profile (*Mayer et al., 2021*). In support of this, previous work using diffusion tensor imaging suggests that MetS-related microstructural white matter alterations preferentially occur in the frontal and temporal lobe, which spatially matches the frontotemporal morphometric differences observed in our work (*Segura*

*et al., 2009*). Furthermore, we speculate on an interplay between local and network-topological susceptibility in MetS: functional and structural connectivity may provide a scaffold for propagating MetS-related perturbation across the network in the sense of a spreading phenomenon – i.e., a region might be influenced by network-driven exposure to regions with higher local susceptibility. Observed degeneration of a region might be aggravated by malfunctional communication to other vulnerable regions including mechanisms of excitotoxicity, diminished excitation and metabolic stress (*Saxena and Caroni, 2011*). These findings underscore the relevance of brain network organization in understanding the pathomechanistic link of MetS and brain morphology.

While this work's strengths lie in a large sample size, high-quality MRI and clinical data, robust image processing, and a comprehensive methodology for examining the link of MetS and brain health, it also has limitations. First, the virtual histology analysis relies on post-mortem brain samples, potentially different from in-vivo profiles. In addition, the predominance of UKB subjects may bias the results, and potential reliability issues of the cognitive assessment in the UKB need to be acknowledged (*Gell et al., 2023*). Lastly, the cross-sectional design restricts the ability for demonstrating causative effects. Longitudinal assessment of the surveyed relationships would provide more robust evidence and therefore, future studies should move in this direction.

## Conclusion

Our analysis revealed associative effects of MetS, structural brain integrity, and cognition, complementing existing efforts to motivate and inform strategies for cardiometabolic risk reduction. In conjunction, a characteristic and reproducible structural imaging fingerprint associated with MetS was identified. This pattern of MetS-related brain morphological abnormalities was linked to local histological as well as global network topological features. Collectively, our results highlight how an integrative, multi-modal, and multi-scale analysis approach can lead to a more holistic understanding of the neural underpinnings of MetS and its risk components. As research in this field advances, leveraging neuroimaging may improve personalized cardiometabolic risk mitigation approaches.

## Materials and methods

### Study population – the UK Biobank and Hamburg City Health Study

Here, we investigated cross-sectional clinical and imaging data from two large-scale population-based cohort studies: (1) the UK Biobank (UKB, n=39,668, age 45–80 years; application number 41655) and (2) the Hamburg City Health Study (HCHS, n=2637, age 45–74 years) (*Miller et al., 2016*; *Jagodzinski et al., 2020*). Both studies recruit large study samples with neuroimaging data alongside a detailed demographic and clinical assessment. Respectively, data for the first visit including a neuroimaging assessment were included. Individuals were excluded if they had a history or a current diagnosis of neurological or psychiatric disease. Field IDs of the used UKB variables are presented in *Appendix 1—table 5*. UKB individuals were excluded based on the non-cancer illnesses codes (http://biobank.ndph.ox.ac.uk/showcase/coding.cgi?id=6). Excluded conditions were Alzheimer's disease; alcohol, opioid, and other dependencies; amyotrophic lateral sclerosis; brain injury; brain abscess; chronic neurological problem; encephalitis; epilepsy; hemorrhage; head injury; meningitis; multiple sclerosis; Parkinson's disease; skull fracture. Same criteria were applied to HCHS individuals based on the neuroradiological evaluation and self-reported diagnoses variables. To enhance comparability to previous studies we supplemented a case-control analysis enabling to complement continuous multivariate statistical analyses by group statistics. Therefore, a MetS sample was identified based on the consensus definition of the International Diabetes Federation (*Appendix 1—table 6*) and matched to a control cohort.

### Ethics approval

The UKB was ethically approved by the North West Multi-Centre Research Ethics Committee (MREC). Details on the UKB Ethics and Governance framework are provided online (https://www.ukbiobank.ac.uk/media/0xsbmfmw/egf.pdf). The HCHS was approved by the local ethics committee of the Landesärztekammer Hamburg (State of Hamburg Chamber of Medical Practitioners, PV5131). Good Clinical Practice (GCP), Good Epidemiological Practice (GEP), and the Declaration of Helsinki were the

ethical guidelines that governed the conduct of the HCHS (*Petersen et al., 2020*). Written informed consent was obtained from all participants investigated in this work.

## Clinical assessment

In the UK Biobank, a battery of cognitive tests is administered, most of which represent shortened and computerized versions of established tests aiming for a comprehensive and concise assessment of cognition (*Sudlow et al., 2015*). From this battery, we investigated tests for executive function and processing speed (Reaction Time Test, Symbol Digit Substitution Test, Tower Rearranging Test, Trail Making Tests A and B), memory (Numeric Memory Test, Paired Associate Learning Test, Prospective Memory Test), and reasoning (Fluid Intelligence Test, Matrix Pattern Completion Test). Detailed descriptions of the individual tests can be found elsewhere (*Fawns-Ritchie and Deary, 2020*). Furthermore, some tests (Matrix Pattern Completion Test, Numeric Memory Test, Paired Associate Learning Test, Symbol Digit Substitution Test, Trail Making Test, and Tower Rearranging Test) are only administered to a subsample of the UKB imaging cohort explaining the missing test results for a subgroup of participants.

In the HCHS, cognitive testing was administered by a trained study nurse and included the Animal Naming Test, Trail Making Test A and B, Multiple Choice Vocabulary Intelligence Test B, and Word List Recall subtests of the Consortium to Establish a Registry for Alzheimer's Disease Neuropsychological Assessment Battery (CERAD-Plus), as well as the Clock Drawing Test (*Morris et al., 1989*; *Shulman, 2000*).

## MRI acquisition

The full UKB neuroimaging protocol can be found online (https://biobank.ctsu.ox.ac.uk/crystal/crystal/docs/brain_mri.pdf; *Miller et al., 2016*). MR images were acquired on a 3 T Siemens Skyra MRI scanner (Siemens, Erlangen, Germany). T1-weighted MRI used a 3D MPRAGE sequence with 1 mm isotropic resolution with the following sequence parameters: repetition time = 2000 ms, echo time = 2.01 ms, 256 axial slices, slice thickness = 1 mm, and in-plane resolution = 1 × 1 mm. In the HCHS, MR images were acquired as well on a 3 T Siemens Skyra MRI scanner. Measurements were performed with a protocol as described in previous work (*Petersen et al., 2020*). In detail, for 3D T1-weighted anatomical images, rapid acquisition gradient-echo sequence (MPRAGE) was used with the following sequence parameters: repetition time = 2500 ms, echo time = 2.12 ms, 256 axial slices, slice thickness = 0.94 mm, and in-plane resolution = 0.83 × 0.83 mm.

## Estimation of brain morphological measures

To achieve comparability and reproducibility, the preconfigured and containerized CAT12 pipeline (CAT12.7 r1743; https://github.com/m-wierzba/cat-container; *Wierzba and Hoffstaedter, 2022*) was employed for surface reconstruction and cortical thickness measurement building upon a projection-based thickness estimation method as well as computation of subcortical volumes (*Gaser et al., 2022*). Cortical thickness measures were normalized from individual to 32 k fsLR surface space (conte69) to ensure vertex correspondence across subjects. Subcortical volumes were computed for the Melbourne Subcortex Atlas parcellation resolution 1 (*Tian et al., 2020*). Volumetric measures for the anterior and posterior thalamus parcels were averaged to obtain a single measure for the thalamus. Individuals with a CAT12 image quality rating lower than 75% were excluded during the quality assessment. To facilitate large-scale data management while ensuring provenance tracking and reproducibility, we employed the DataLad-based FAIRly big workflow for image data processing (*Wagner et al., 2022*).

## Statistical analysis

Statistical computations and plotting were performed in python 3.9.7 leveraging bctpy (v. 0.6.0), brainstat (v. 0.3.6), brainSMASH (v. 0.11.0), and the ENIGMA toolbox (v. 1.1.3). matplotlib (v. 3.5.1), neuromaps (v. 0.0.1), numpy (v. 1.22.3), pandas (v. 1.4.2), pingouin (v. 0.5.1), pyls (v. 0.0.1), scikit-learn (v. 1.0.2), scipy (v. 1.7.3), seaborn (v. 0.11.2) as well as in matlab (v. 2021b) using ABAnnotate (v. 0.1.1).

### Partial least squares correlation analysis

To relate MetS components and brain morphology, we performed a PLS using pyls (https://github.com/rmarkello/pyls; *Markello, 2021*). PLS identifies covariance profiles that relate to two sets of variables in a data-driven multivariate analysis (*Krishnan et al., 2011*). Here, we related regional cortical

thickness and subcortical volumes to clinical measurements of MetS components, i.e., obesity (waist circumference, hip circumference, waist-hip ratio, body mass index), arterial hypertension (systolic blood pressure, diastolic blood pressure), dyslipidemia (high-density lipoprotein, low-density lipoprotein, total cholesterol, triglycerides) and insulin resistance (HbA1c, non-fasting blood glucose). Before conducting the PLS, missing values were imputed via k-nearest neighbor imputation ($n_{neighbor}$ = 4) with imputation only taking into account variables of the same group, i.e., MetS component variables were imputed based on the remaining MetS component data only and not based on demographic variables. To account for age, sex, education, and cohort (UKB/HCHS) as potential confounds, they were regressed out of brain morphological and MetS component data.

We then performed PLS as described in previous work (*Petersen et al., 2022b*). Methodological details are covered in *Figure 1a* and *Box 1*. Brain morphological measures were randomly permuted ($n_{permute}$ = 5000) to assess the statistical significance of derived latent variables and their corresponding covariance profiles. Subject-specific PLS scores, including a clinical score and an imaging score, were computed. Higher scores indicate stronger adherence to the respective covariance profiles: a high clinical score signifies pronounced expression of the clinical profile, and a high imaging score reflects marked adherence to the brain morphological profile. Bootstrap resampling ($n_{bootstrap}$ = 5000) was performed to assess the contribution of individual variables to the imaging-clinical relationship. Confidence intervals (95%) of singular vector weights were computed for clinical variables to assess the significance of their contribution. To estimate the contributions of brain regions, bootstrap ratios were computed as the singular vector weight divided by the bootstrap-estimated standard error. A high bootstrap ratio is indicative of a region's contribution, as a relevant region shows a high singular vector weight alongside a small standard error implying stability across bootstraps. The bootstrap ratio equals a z-score in the case of a normally distributed bootstrap. Hence, brain region contributions were considered significant if the bootstrap ratio was >1.96 or <−1.96 (95% confidence interval). Overall model robustness was assessed via a 10-fold cross-validation by correlating out-of-sample PLS scores within each fold.

## Mediation analysis

In a post-hoc mediation analysis, we investigated how the subject-specific clinical PLS score of the first latent variable, reflecting the degree of an individual's expression of the identified MetS risk profile, relates to cognitive test outcomes, and whether this relationship is influenced by the imaging PLS score of the first latent variable, which represents the degree of brain morphological differences (*Figure 1b*). This analysis allows to separate the total effect of the clinical PLS score on cognitive performance into: (1) a direct effect (the immediate link of clinical scores and cognition), and (2) an indirect effect (the portion influenced by the imaging PLS score). This approach helps to disentangle the complex interplay between MetS and cognitive function by examining the role of brain structural effects as a potential intermediary. We considered an indirect effect as mediating if there was a significant association between the clinical and imaging PLS scores, the imaging PLS score was significantly associated with the cognitive outcome, and if the link between clinical scores and cognitive outcomes weakened (partial mediation) or became insignificant (full mediation) after accounting for imaging scores. The significance of mediation was assessed using bootstrapping ($n_{bootstrap}$ = 5000), with models adjusted for age, sex, and education. To obtain standardized estimates, mediation analysis inputs were z-scored beforehand. Given the variation in cognitive test batteries between the UKB and HCHS cohorts, only individuals with results from the respective tests were considered in each mediation analysis. To account for the different versions of the Trail Making Tests A and B used in both cohorts, test results were harmonized through z-scoring within the individual subsamples before a pooled z-scoring step.

## Contextualization analysis

We investigated the link of MetS and regional brain morphological measurements in the context of cell-specific gene expression profiles and structural and functional brain network characteristics (*Figure 1c*). Therefore, we used the Schaefer-parcellated (400×7 and 100×7, v.1) bootstrap ratio map and related it to indices representing different gene expression and network topological properties of the human cortex via spatial correlations (Spearman correlation, $r_{sp}$) on a group-level (*Schaefer et al., 2018*).

## Box 1. Partial least squares correlation analysis explained.

Regional morphometric information (Schaefer 400- and Melbourne Subcortical Atlas-parcellated) and clinical data (age sex, education, and MetS component data) were arranged in two matrices $X_{n_{participants} \times n_{brain\ regions}}$ and $Y_{n_{participants} \times n_{clinical\ variables}}$ and then z-scored. Subsequently, a clinical-anatomical correlation matrix was calculated. Singular value decomposition was performed on the correlation matrix which resulted in a set of mutually orthogonal latent variables. The smaller dimension of the correlation matrix – its rank – equals the latent variable count. In our case, this was the number of clinical variables. Singular value decomposition results in a left singular vector matrix ($U_{n_{brain\ regions} \times n_{latent\ variables}}$), right singular vector matrix ($V_{n_{clinical\ variables} \times n_{latent\ variables}}$) and a diagonal matrix of singular values ($\Delta_{n_{latent\ variables} \times n_{latent\ variables}}$). Together, these represent a set of latent variables with a latent variable being composed of a left and right singular vector and a corresponding singular value. Each latent variable represents a specific covariance profile within the input data. A singular vector weights the corresponding original variables to maximize their covariance, i.e., the weighted regional values of a singular vector $U_{brain\ regions,\ latent\ variable\ j}$ can be interpreted as a maximally covarying brain morphology pattern and its corresponding clinical substrate ($V_{clinical\ variables,\ latent\ variable\ j}$). The explained variance of a latent variable was calculated as the ratio of its corresponding squared singular value to the sum of the remaining squared singular values. Significance of a latent variable was assessed by comparing the observed explained variance to a non-parametric distribution of permuted values acquired by permuting the subject order in $X$ ($n_{permute} = 5000$).

Subject-specific PLS scores measure to which extent an individual expresses a covariance profile represented by a latent variable. Thus, scores can be thought of as factor weightings in factor analysis. A high score describes the high agreement of a participant with the identified pattern. They were calculated by projecting $U$ on $X$ for an imaging score

$$Imaging\ score = UX$$

and $V$ on $Y$ for a clinical score

$$Clinical\ score = VY$$

Bootstrap resampling was performed to identify brain regions and clinical variables with a high and robust contribution to the clinical-anatomical association. Individuals were randomly sampled from $X$ and $Y$ with replacement (n=5000) which resulted in a set of resampled correlation matrices propagated to singular value decomposition resulting in a sampling distribution of singular vector weights for each input variable. This enabled the computation of 95% confidence intervals for the clinical variables and a bootstrap ratio for the brain regions.

$$Bootstrap\ ratio = \frac{Singular\ vector\ weight\ U_{brain\ region\ i,\ latent\ variable\ j}}{Standard\ error\ estimated\ from\ bootstrapping}$$

The bootstrap ratio measures a brain region's contribution to the observed covariance profile of a respective latent variable, as a relevant region shows a high singular vector weight alongside a small standard error implying stability across bootstraps.

Virtual histology analysis. We performed a virtual histology analysis leveraging gene transcription information to quantify the density of different cell populations across the cortex employing the ABAnnotate toolbox (**Lotter, 2022**; **Dukart et al., 2021**). Genes corresponding with specific cell populations of the central nervous system were identified based on a classification derived from single nucleus-RNA sequencing data (**Lake et al., 2016**). The gene-cell type mapping is provided by the

PsychENCODE database (http://resource.psychencode.org/Datasets/Derived/SC_Decomp/DER-19_Single_cell_markergenes_TPM.xlsx; *Wang et al., 2018*). The abagen toolbox (v. 0.1.3) was used to obtain regional microarray expression data of these genes for Schaefer 100×7 parcels based on the Allen Human Brain Atlas (AHBA) (*Markello et al., 2021*). The Schaefer 100×7 atlas was used as it better matches the sampling density of the AHBA eventually resulting in no parcels with missing values. Regional expression patterns of genes corresponding to astrocytes, endothelial cells, excitatory neuron populations (Ex1-8), inhibitory neuron populations (In1-8), microglia, and oligodendrocytes were extracted. Instead of assessing the correspondence between MetS effects and the expression pattern of each gene directly, we employed ensemble-based gene category enrichment analysis (GCEA) (*Fulcher et al., 2021*). This approach represents a modification to customary GCEA addressing the issues of gene-gene dependency through within-category co-expression which is caused by shared spatial embedding as well as spatial autocorrelation of cortical transcriptomics data. In brief, gene transcription indices were averaged within categories (here cell populations) and spatially correlated with the bootstrap ratio map. Statistical significance was assessed by comparing the empirical correlation coefficients against a null distribution derived from surrogate maps with preserved spatial embedding and autocorrelation computed via a spatial lag model (*Burt et al., 2018*). Further details on the processing steps covered by ABAnnotate can be found elsewhere (https://osf.io/gcxun; *Lotter et al., 2023*).

Brain network topology. To investigate the cortical MetS effects pattern in the context of brain network topology, three connectivity metrics were leveraged based on data from structural and functional brain imaging: weighted degree centrality, neighborhood abnormality as well as macroscale functional connectivity gradients as described previously (*Petersen et al., 2022b*). These were computed based on functional and structural consensus connectomes at group-level derived from the Human Connectome Project Young Adults dataset comprised in the ENIGMA toolbox (*Larivière et al., 2021*; *Larivière et al., 2020*). The preprocessing of these connectomes is described elsewhere (*Larivière et al., 2020*).

Weighted degree centrality. Weighted degree centrality is a measure of a brain region's topological relevance and is commonly used for the identification of brain network hubs (*van den Heuvel and Sporns, 2013*). The degree centrality of a node $i$ was computed as the sum of its functional or structural connection weights (*Rubinov and Sporns, 2010*). The resulting values were ranked before further analysis.

Neighborhood abnormality. Neighborhood abnormality represents a summary measure of a cortical property in the node neighborhood defined by functional or structural brain network connectivity (*Shafiei et al., 2020*). In this work, the MetS-related morphological abnormalities (bootstrap ratio or t-statistic) in nodes $j$ connected to node $i$ were averaged and weighted by their respective functional or structural seed connectivity ($w_{ij}$):

$$A_i = \frac{1}{N_i} \sum_{j \in N_i} C_j w_{ij}$$

where $j$ is one of the connected nodes $N_i$, $C_j$ is the measure of MetS-related effects on cortical thickness and the corresponding connection weight $w_{ij}$. The term $\frac{1}{N_i}$ corrects for the nodal degree by normalizing the number of connections. For example, a high positive or negative $A_i$ represents strong connectivity to nodes of pronounced MetS effects (*Petersen et al., 2022b*).

Functional connectivity gradients. To contextualize the MetS-related morphological abnormalities with the functional network hierarchy, we derived macroscale functional connectivity gradients as a proxy of the canonical sensorimotor-association axis, which determines the distribution of manifold cortical properties (*Margulies et al., 2016*; *Sydnor et al., 2021*). Functional connectivity gradients were derived by applying diffusion map embedding on the HCP functional connectivity matrix using BrainSpace (*Vos de Wael et al., 2020*). A functional connectivity gradient can be interpreted as a spatial axis of connectivity variation spanning the cortical surface, as nodes of similar connectivity profiles are closely located on these axes.

For this analysis, the statistical significance of spatial correlations was assessed via spin permutations (n=1000) which represent a null model preserving the inherent spatial autocorrelation of cortical information (*Alexander-Bloch et al., 2018*). Spin permutations are performed by projecting parcelwise data onto a sphere which then is randomly rotated. After rotation, information is projected back

on the surface, and a permuted $r_{sp}$ is computed. A p-value is computed comparing the empirical correlation coefficient to the permuted distribution. To assure that our results do not depend on null model choice, we additionally tested our results against a variogram-based null model implemented in the brainSMASH toolbox (https://github.com/murraylab/brainsmash; *Burt and Murray, 2020*) as well as a network rewiring null model with preserved density and degree sequence (*Burt et al., 2020*; *Maslov et al., 2004*).

All p-values resulting from both contextualization analyses were FDR-corrected for multiple comparisons. As we conducted this study mindful of the reuse of our resources, the MetS effect maps are provided as separate supplementary files to enable further analyses (*Supplementary files 1-3*).

### Sensitivity analyses

For a sensitivity analysis, we reperformed the PLS separately within the UKB and HCHS cohorts. In contrast to the PLS main analysis, in these subset-specific PLS analyses cognitive test performances were also incorporated as clinical variables as cognitive batteries were subset-specific. This approach was employed to evaluate the stability of the results and to determine if cognitive tests contribute to the latent variables.

To test whether the PLS indeed captures the link of MetS and brain morphology, we conducted a group comparison as in previous studies of MetS. Besides descriptive group statistics, the cortical thickness of individuals with MetS and matched controls was compared on a surface vertex-level leveraging the BrainStat toolbox (v 0.3.6, https://brainstat.readthedocs.io/) (*Larivière et al., 2023*). A general linear model was applied correcting for age, sex, education, and cohort effects. Vertex-wise p-values were FDR-corrected for multiple comparisons. To demonstrate the correspondence between the t-statistic and cortical bootstrap ratio maps, we related them via spatial correlation analyses. The t-statistic map was also used for sensitivity analysis of the virtual histology analysis and brain network contextualization.

To ensure that the brain network contextualization results were not biased by the connectome choice, we reperformed the analysis with structural and functional group consensus connectomes based on resting-state functional and diffusion-weighted MRI data from the HCHS. The corresponding connectome reconstruction approaches were described elsewhere (*Petersen et al., 2022b*).

## Acknowledgements

German Research Foundation, Schwerpunktprogramm (SPP) 2041, (grant number 454012190) to SBE, GT; German Research Foundation, Sonderforschungsbereich (SFB) 936 (grant number 178316478) project C2 - MP, CM, JF, GT and BC and C7 - SK, JG; German Research Foundation, SFB 1451 & IRTG 2150 to SBE; National Institute of Health, R01 (grant number MH074457) to SBE; European Union's Horizon 2020 Research and Innovation Program (grant number 945539, 826421) to SBE; German Center for Cardiovascular Research (FKZ 81Z1710101 and FKZ 81Z0710102); the Hamburg City Health Study is also supported by Amgen, Astra Zeneca, Bayer, BASF, Deutsche Gesetzliche Unfallversicherung (DGUV), Deutsches Institut für Ernährungsforschung, the Innovative medicine initiative (IMI) under grant number No. 116074, the Fondation Leducq under grant number 16 CVD 03., the euCanSHare grant agreement under grant number 825903-euCanSHare H2020, Novartis, Pfizer, Schiller, Siemens, Unilever, and 'Förderverein zur Förderung der HCHS e.V.'

## Additional information

### Competing interests

Tanja Zeller: TZ is listed as co-inventor of an international patent on the use of a computing device to estimate the probability of myocardial infarction (PCT/EP2021/073193, International Publication Number WO2022043229A1). TZ is shareholder of the company ART-EMIS GmbH Hamburg. Jürgen Gallinat: JG has received speaker fees from Lundbeck, Janssen-Cilag, Lilly, Otsuka and Boehringer outside the submitted work. Jens Fiehler: JF reported receiving personal fees from Acandis, Cerenovus, Microvention, Medtronic, Phenox, and Penumbra; receiving grants from Stryker and Route 92; being managing director of eppdata; and owning shares in Tegus and Vastrax; all outside the

submitted work. Raphael Twerenbold: RT is listed as co-inventor of an international patent on the use of a computing device to estimate the probability of myocardial infarction (PCT/EP2021/073193, International Publication Number WO2022043229A1). RT is shareholder of the company ART-EMIS GmbH Hamburg. Goetz Thomalla: GT has received fees as consultant or lecturer from Acandis, Alexion, Amarin, Bayer, Boehringer Ingelheim, BristolMyersSquibb/Pfizer, Daichi Sankyo, Portola, and Stryker outside the submitted work. The other authors declare that no competing interests exist.

## Funding

| Funder | Grant reference number | Author |
| --- | --- | --- |
| German Research Foundation | 454012190 | Simon B Eickhoff Goetz Thomalla |
| German Research Foundation | 178316478 | Bastian Cheng |
| National Institutes of Health | MH074457 | Simon B Eickhoff |
| Horizon 2020 - Research and Innovation Framework Programme | 945539 | Simon B Eickhoff |
| Horizon 2020 - Research and Innovation Framework Programme | 826421 | Simon B Eickhoff |

The funders had no role in study design, data collection and interpretation, or the decision to submit the work for publication.

## Author contributions

Marvin Petersen, Conceptualization, Resources, Data curation, Software, Formal analysis, Investigation, Visualization, Methodology, Writing - original draft, Project administration, Writing – review and editing; Felix Hoffstaedter, Conceptualization, Resources, Data curation, Software, Writing – review and editing; Felix L Nägele, Carola Mayer, Maximilian Schell, Data curation, Writing – review and editing; D Leander Rimmele, Birgit-Christiane Zyriax, Tanja Zeller, Simone Kühn, Jürgen Gallinat, Jens Fiehler, Raphael Twerenbold, Amir Omidvarnia, Kaustubh R Patil, Writing – review and editing; Simon B Eickhoff, Resources, Funding acquisition, Writing – review and editing; Goetz Thomalla, Supervision, Funding acquisition, Writing – review and editing; Bastian Cheng, Conceptualization, Resources, Supervision, Funding acquisition, Writing – review and editing

## Author ORCIDs

Marvin Petersen ⬤ http://orcid.org/0000-0001-6426-7167
Felix Hoffstaedter ⬤ https://orcid.org/0000-0001-7163-3110
Carola Mayer ⬤ http://orcid.org/0000-0002-8065-8683
Raphael Twerenbold ⬤ https://orcid.org/0000-0003-3814-6542
Amir Omidvarnia ⬤ http://orcid.org/0000-0002-4744-8721

## Ethics

The UKB was ethically approved by the North West Multi-Centre Research Ethics Committee (MREC). Details on the UKB Ethics and Governance framework are provided online (https://www.ukbiobank.ac.uk/media/0xsbmfmw/egf.pdf). The HCHS was approved by the local ethics committee of the Landesärztekammer Hamburg (State of Hamburg Chamber of Medical Practitioners, PV5131). Good Clinical Practice (GCP), Good Epidemiological Practice (GEP) and the Declaration of Helsinki were the ethical guidelines that governed the conduct of the HCHS (Petersen et al., 2020). Written informed consent was obtained from all participants investigated in this work.

Reviewer #1 (Public Review): https://doi.org/10.7554/eLife.93246.3.sa1
Reviewer #2 (Public Review): https://doi.org/10.7554/eLife.93246.3.sa2
Author Response https://doi.org/10.7554/eLife.93246.3.sa3

## Additional files

### Supplementary files

• Supplementary file 1. Schaefer 400×7-parcellated maps of metabolic syndrome (MetS)-related brain morphological abnormalities (bootstrap ratio from PLS, t-statistic from group comparison).

• Supplementary file 2. Schaefer 100×7-parcellated bootstrap ratio map.

• Supplementary file 3. t-statistic from group comparison on fsLR.

• MDAR checklist

### Data availability

UK Biobank data can be obtained via its standardized data access procedure (https://www.ukbiobank.ac.uk/). HCHS participant data used in this analysis is not publicly available for privacy reasons, but access can be established via request to the HCHS steering committee. The analysis code is publicly available on GitHub (https://github.com/csi-hamburg/2023_petersen_mets_brain_morphology (copy archived at *Petersen, 2023*) and https://github.com/csi-hamburg/CSIframe/wiki/Structural-processing-with-CAT).

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

# Appendix 1

## General appendix

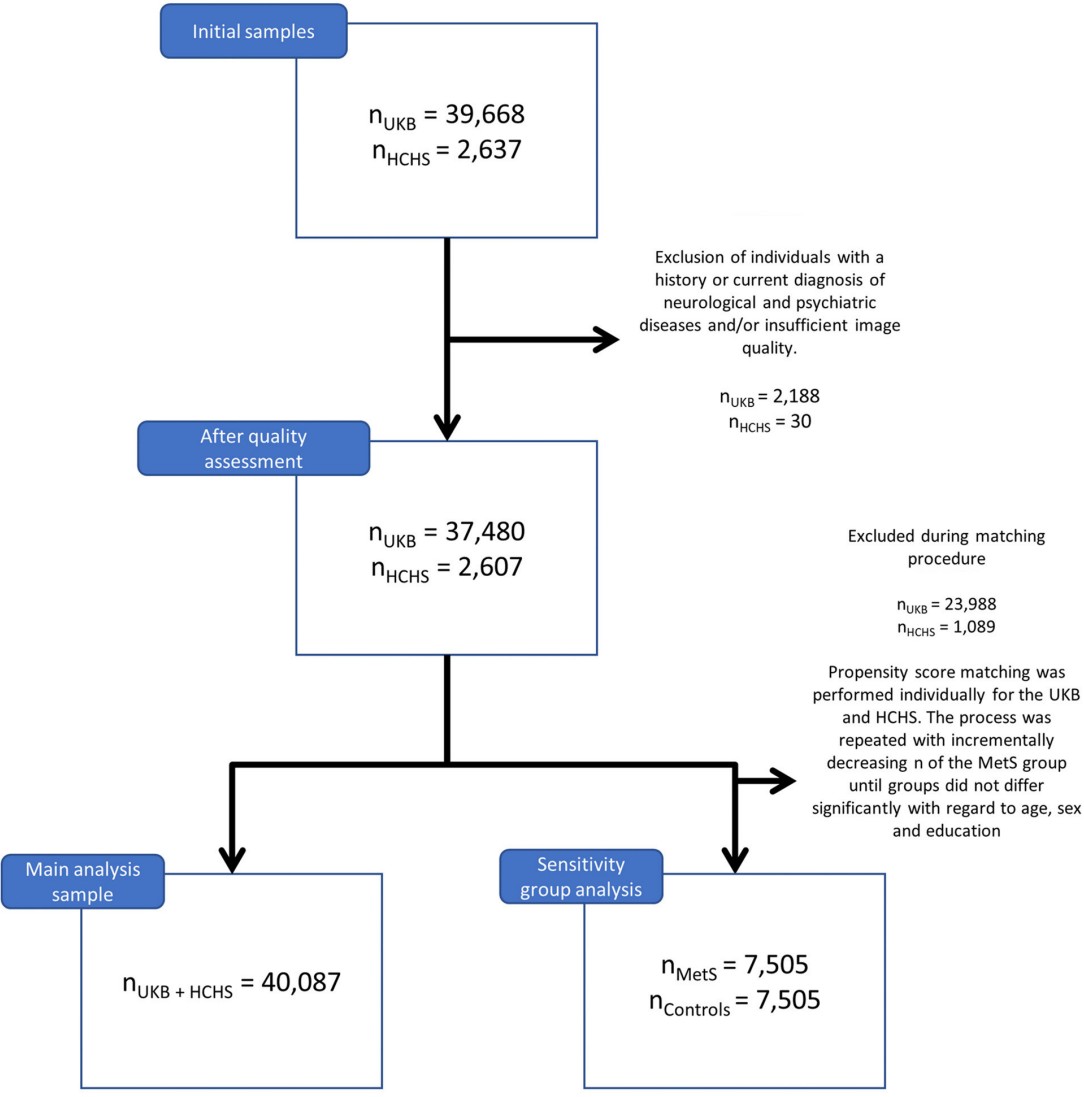

**Appendix 1—figure 1.** Flowchart sample selection procedure.

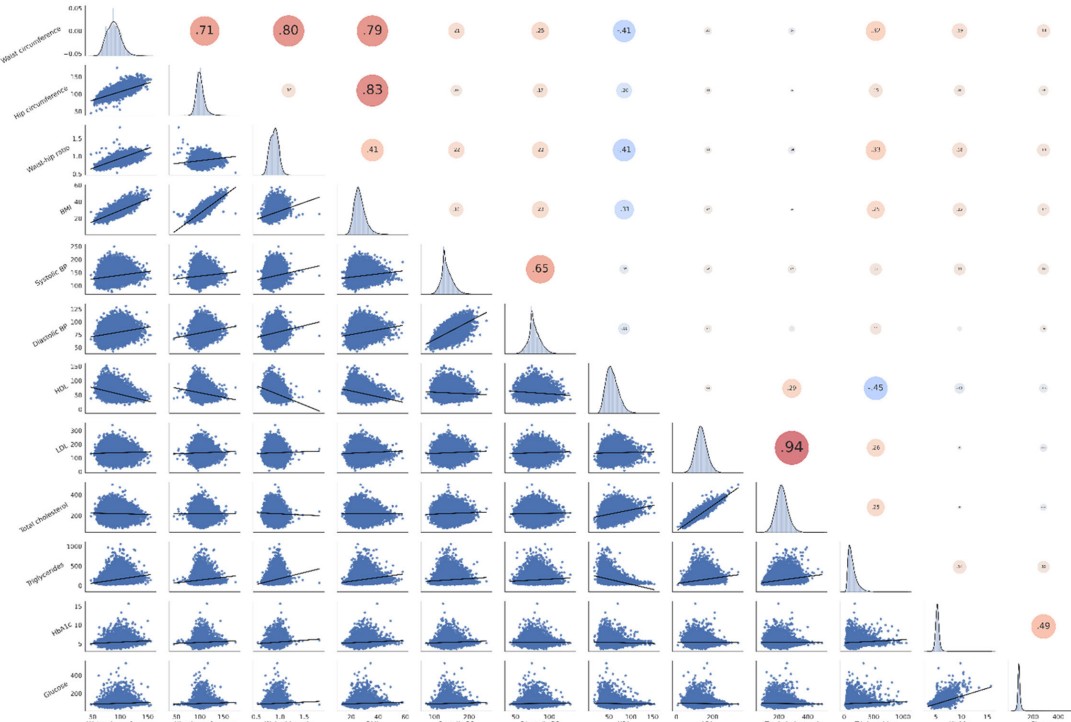

**Appendix 1—figure 2.** Correlation matrix of metabolic syndrome-related risk factors. The upper triangle of the matrix displays Pearson correlations with dot size and color representing the magnitude of the coefficients. The diagonal shows kernel density plots. The lower triangle illustrates the variables' linear relationships via regression plots. Of note, non fasting plasma glucose was investigated in this analysis. Abbreviations: BP – blood pressure.

**Appendix 1—table 1.** Partial least squares analysis - latent variables.

| Latent variable | Explained variance (%) | p-value |
| --- | --- | --- |
| 0 | 71.20 | 0.0002 |
| 1 | 22.33 | 0.0002 |
| 2 | 2.12 | 0.0002 |
| 3 | 1.84 | 0.0006 |
| 4 | 1.03 | 0.0026 |
| 5 | 0.52 | 0.0266 |
| 6 | 0.38 | 0.0100 |
| 7 | 0.23 | 0.0032 |
| 8 | 0.18 | 0.1178 |
| 9 | 0.16 | 0.2122 |
| 10 | 0.00 | 0.3137 |
| 11 | 0.00 | 0.0608 |
| 12 | 0.00 | 1 |
| 13 | 0.00 | 1 |
| 14 | 0.00 | 1 |
| 15 | 0.00 | 1 |

**Appendix 1—table 2.** Partial least squares analysis-Cross-validation.

| CV fold | $r_{sp}$ |
|---|---|
| 0 | 0.17 |
| 1 | 0.21 |
| 2 | 0.22 |
| 3 | 0.16 |
| 4 | 0.15 |
| 5 | 0.18 |
| 6 | 0.23 |
| 7 | 0.13 |
| 8 | 0.20 |
| 9 | 0.22 |

**Appendix 1—table 3.** Virtual histology analysis - Bootstrap ratio (partial least squares, PLS).

| Cell type | $Zr_{sp}$ | $p_{FDR}$ |
|---|---|---|
| Endo | 0.190 | 0.016 |
| Micro | 0.271 | 0.016 |
| Ex8 | 0.165 | 0.016 |
| In1 | 0.363 | 0.036 |
| Ex6 | 0.146 | 0.034 |
| Oligo | 0.207 | 0.057 |
| In7 | 0.079 | 0.083 |
| Ex1 | 0.122 | 0.144 |
| In2 | 0.058 | 0.179 |
| In3 | 0.047 | 0.208 |
| Astro | 0.071 | 0.259 |
| In8 | 0.055 | 0.299 |
| Ex7 | 0.044 | 0.336 |
| In5 | 0.037 | 0.388 |
| Ex4 | −0.020 | 0.776 |
| Ex5 | −0.055 | 0.924 |
| In4 | −0.056 | 0.949 |
| In6 | −0.099 | 0.949 |
| Ex2 | −0.102 | 0.967 |
| Ex3 | −0.289 | 0.999 |

**Appendix 1—table 4.** Virtual histology analysis - t-statistic (group comparison).

| Cell type | $Zr_{sp}$ | $p_{FDR}$ |
|---|---|---|
| Endo | 0.208 | 0.020 |
| Micro | 0.321 | 0.040 |
| Ex8 | 0.208 | 0.040 |
| Oligo | 0.233 | 0.055 |

*Appendix 1—table 4 Continued on next page*

*Appendix 1—table 4 Continued*

| Cell type | Zr$_{sp}$ | p$_{FDR}$ |
|---|---|---|
| In1 | 0.432 | 0.108 |
| Ex6 | 0.145 | 0.123 |
| Ex1 | 0.156 | 0.229 |
| In3 | 0.058 | 0.233 |
| Astro | 0.120 | 0.233 |
| In7 | 0.059 | 0.233 |
| In2 | 0.063 | 0.233 |
| Ex7 | 0.089 | 0.263 |
| In5 | 0.063 | 0.300 |
| In8 | 0.066 | 0.317 |
| Ex4 | 0.015 | 0.585 |
| Ex5 | −0.007 | 0.690 |
| In6 | −0.078 | 0.861 |
| Ex2 | −0.070 | 0.861 |
| In4 | −0.087 | 0.901 |
| Ex3 | −0.341 | 0.997 |

**Appendix 1—table 5.** UK Biobank field IDs.

| **Age** | **21003** |
|---|---|
| Sex | 31 |
| Education | 6133* |
| Waist circumference | 48 |
| Hip circumference | 49 |
| Body mass index | 21001 |
| RR$_{systolic}$ | 4080 |
| RR$_{diastolic}$ | 4079 |
| HDL | 30760 |
| LDL | 30780 |
| Cholesterol | 30690 |
| Triglycerides | 30870 |
| HbA1c | 30750 |
| Blood glucose | 30740 |
| Medication for cholesterol, blood pressure, diabetes | 6153 |
| Fluid Intelligence | 20191 |
| Matrix Pattern Completion | 6373 |
| Numeric Memory Test | 20240 |
| Paired Associate Learning | 20197 |
| Prospective Memory | 20018 |
| Reaction Time | 20023 |

*Appendix 1—table 5 Continued on next page*

*Appendix 1—table 5 Continued*

| Age | 21003 |
| --- | --- |
| Symbol Digit Substitution | 20159 |
| Tower Rearranging Test | 21004 |
| Trail Making Test A | 6348 |
| Trail Making Test B | 6350 |

Abbreviations: RR = blood pressure.

*Converted to International Standard Classification of Education (ISCED) via the UKBB parser (https://github.com/USC-IGC/ukbb_parser; *Zhu et al., 2019*).

**Appendix 1—table 6.** Metabolic syndrome Criteria of the International Diabetes Federation (IDF) (*Alberti et al., 2006*).

**Metabolic syndrome = obesity + two further criteria**

| | |
| --- | --- |
| Obesity | waist circumference ♀:≥80 cm; ♂:≥94 cm |
| Dyslipidemia (raised triglycerides) | ≥150 mg/dL (1.7 mmol/L) or lipid lowering medication |
| Dyslipidemia (reduced HDL cholesterol) | ♀:<50 mg/dL (1.29 mmol/L); ♂:<40 mg/dL (1.03 mmol/L) in males |
| Arterial hypertension (raised blood pressure) | systolic BP ≥130 or diastolic BP ≥85 mm Hg or antihypertensive medication or diagnosis of hypertension |
| Insuline resistance | Fasting plasma glucose ≥100 mg/dL (5.6 mmol/L) or antidiabetic therapy or diagnosis of diabetes mellitus type 2* |

*Measurements of fasting plasma glucose were not available for the study sample. Consequently, the criterion of insulin resistance was only based on the diagnosis of diabetes mellitus and administration of antidiabetic therapy.

## Appendix 2

### Case-control analysis

As a sensitivity analysis and to facilitate the comparison with previous reports which mainly rely on group statistics, we supplemented the continuous partial least squares correlation analysis with a group analysis based on a case-control design.

### Matching procedure

After quality assessment, individuals with metabolic syndrome were identified based on the consensus criteria of the *International Diabetes Federation* ($n_{UKB}$ = 6746, $n_{HCHS}$ = 759). An individual was considered to exhibit MetS in case of obesity (increased waist circumference) and two further criteria being raised plasma triglycerides, reduced HDL cholesterol, arterial hypertension or insulin resistance. Of note, measurements of fasting plasma glucose were not available for the study sample. Consequently, the criterion of insulin resistance was only based on the diagnosis of diabetes mellitus and administration of antidiabetic therapy. Within each cohort, an equally sized control cohort was sampled which was matched for age, sex and education (International Standard Classification of Education) using propensity score matching as implemented in the matchit (v4.3.3) R package. MetS and control samples from both cohorts were pooled yielding an analysis sample of 15,010 individuals ($n_{MetS}$ = 7505,, $n_{controls}$ = 7505). For detailed matching results refer to *Appendix 2—figures 1 and 2* shown below.

### Group comparison of clinical data

Sample characteristics were compared between participants with MetS and controls using $\chi^2$-tests for binary and two-sample t-tests for continuous data. Cognitive variables were compared within UKB and HCHS subgroups via analyses of covariance (ANCOVA) adjusting for age, sex and education. Resulting test statistics were converted to Cohen's d which quantifies the group difference in standard deviations. P-values were false-discovery rate (FDR)-corrected for multiple comparisons. Separate group statistics of demographic, risk and cognitive variables for the UKB and HCHS are shown in *Appendix 1—table 1–2*. Individuals with MetS exhibited a more severe risk profile indicating that the group definitions captured considerable differences in the MetS components profile. Group differences regarding MetS criteria proportions are visualized in *Appendix 2—figure 3*. As the cognitive assessment of the UKB and HCHS differed, cognitive scores were compared between groups within the individual studies. UKB subjects with MetS performed significantly worse in the Fluid Intelligence Test (6.66±2.10 vs 6.82±2.09, Cohen's d=0.08, $p_{FDR}$ < 0.001), Numeric Memory Test (6.64±1.61 vs 6.84±1.53, Cohen's d=.12, $p_{FDR}$ < 0.001), Paired Associate Learning Test (6.45±2.60 vs 6.73±2.61, Cohen's d=0.10, $p_{FDR}$ < 0.001) and Symbol Digit Substitution Test (18.47±5.12 vs 19.00±5.16, Cohen's d=0.10, $p_{FDR}$ < 0.001). HCHS subjects exhibiting MetS showed worse cognitive performance in the Animal Naming Test (23.71±6.46 vs 24.77±6.75, Cohen's d=0.16, $p_{FDR}$ < 0.009) and Multiple-choice Vocabulary Intelligence Test (31.18±3.43 vs 31.71±3.22, Cohen's d=0.16, $p_{FDR}$ < 0.034).

### Vertex-wise cortical thickness analysis

The cortical thickness of individuals with MetS and matched controls were compared on a surface vertex-level leveraging the BrainStat toolbox (v 0.3.6, https://brainstat.readthedocs.io/). Corresponding results are shown in *Appendix 2—figure 4*. The vertex-wise t-statistic, which captures the differential MetS effects across the cortical surface, was Schaefer 400 and Schaefer 100-parcellated and propagated to further analyses. The t-statistic map strongly correlated with the bootstrap ratio maps derived from the PLS analyses. Furthermore, the t-statistic map was significantly associated with density of endothelial cells ($Z_{r_{sp}}$ = 0.208, $p_{FDR}$ = 0.040), microglia ($Z_{r_{sp}}$ = 0.321, $p_{FDR}$ = 0.040), excitatory neurons type 8 ($Z_{r_{sp}}$ = 0.208, $p_{FDR}$ = 0.004) and also correlated significantly with the functional neighborhood abnormality ($r_{sp}$ = 0.313, $p_{spin}$ = 0.024, $p_{smash}$ = 0.018, $p_{rewire}$ < 0.001) and structural neighborhood abnormality ($r_{sp}$ = 0.775, $p_{spin}$ = <0.001, $p_{smash}$ < 0.001, $p_{rewire}$ < 0.001).

**Appendix 2—table 1.** Descriptive group statistics - UK Biobank.

| Metric* | Individuals with MetS | Matched controls | $P_{uncorr}$ | $P_{FDR}$ | Stat[†] |
|---|---|---|---|---|---|
| Age (years) | 64.73±7.42 (6746) | 64.51±7.27 (6746) | 0.095 | 0.154 | −0.03 |
| Sex (% female) | 18.81 (6746) | 18.81 (6746) | >0.99 | >0.99 | 0 |
| Education (ISCED) | 2.63±0.73 (6746) | 2.67±0.71 (6746) | 0.036 | 0.069 | 0.04 |
| | | | | | |
| _Metabolic syndrome criteria_ | | | | | |
| Waist circumference (cm) | 97.39±10.21 (6726) | 88.22±10.59 (6595) | <0.001 | <0.001 [‡] | −0.88 |
| $RR_{systolic}$ (mmHg) | 146.41±15.38 (6213) | 135.57±17.71 (5397) | <0.001 | <0.001 [‡] | −0.66 |
| $RR_{diastolic}$ (mmHg) | 82.26±9.39 (6214) | 77.58±9.79 (5397) | <0.001 | <0.001 [‡] | −0.49 |
| Antihypertensive therapy (%) | 9.96 (6746) | 9.68 (6746) | <0.001 | <0.001 [‡] | 7.07 |
| HDL (mmol/L) | 1.18±0.26 (6225) | 1.49±0.32 (6332) | <0.001 | <0.001 [‡] | 1.08 |
| Triglycerides (mmol/L) | 2.43±1.13 (6617) | 1.30±0.59 (6543) | <0.001 | <0.001 [‡] | −1.25 |
| Lipid-lowering therapy (%) | 39.05 (6746) | 7.07 (6746) | <0.001 | <.001 [‡] | 2446.5 |
| Blood glucose (mmol/L) | 5.18±1.41 (6219) | 4.92±0.68 (6325) | <0.001 | <.001 [‡] | −0.23 |
| Antidiabetic therapy (%) | 0.06 (6746) | 0.19 (6746) | 0.052 | 0.097 | 3.77 |
| | | | | | |
| _Cognitive scores_ | | | | | |
| Fluid Intelligence | 6.66±2.10 (6221) | 6.82±2.09 (6241) | <0.001 | <0.001 [‡] | 0.08 |
| Matrix Pattern Completion | 8.02±2.13 (4283) | 8.14±2.06 (4355) | 0.055 | 0.096 | 0.06 |
| Numeric Memory Test | 6.64±1.61 (4419) | 6.84±1.53 (4505) | <0.001 | <0.001 [‡] | 0.12 |
| Paired Associate Learning | 6.45±2.60 (4337) | 6.73±2.61 (4392) | <0.001 | <0.001 [‡] | 0.10 |
| Prospective Memory | 1.05±0.40 (6362) | 1.06±0.39 (6349) | 0.221 | 0.339 | 0.02 |
| Reaction Time | 590.75±108.27 (6331) | 590.15±111.13 (6325) | 0.792 | 0.858 | −0.005 |

_Appendix 2—table 1 continued on next page_

*Appendix 2—table 1 continued*

| Metric* | Individuals with MetS | Matched controls | $P_{uncorr}$ | $P_{FDR}$ | Stat[†] |
|---|---|---|---|---|---|
| Symbol Digit Substitution | 18.47±5.12 (4292) | 19.00±5.16 (4353) | <0.001 | <0.001 [‡] | 0.10 |
| Tower Rearranging Test | 10.00±3.28 (4255) | 10.08±3.20 (4325) | 0.747 | 0.845 | 0.02 |
| Trail Making Test A (sec) | 226.83±86.26 (4337) | 224.06±83.06 (4392) | 0.643 | 0.836 | −0.03 |
| Trail Making Test B (sec) | 561.81±271.39 (4337) | 553.62±282.55 (4392) | 0.611 | 0.756 | −0.03 |
| **Imaging** | | | | | |
| Mean cortical thickness (mm) | 2.392±0.09 (6746) | 2.397±0.09 (6746) | 0.035 | 0.071 | 0.05 |

Abbreviations: cm = centimeter, dL = deciliter, HDL = high-density lipoprotein, ISCED = International Standard Classification of Education, MetS = metabolic syndrome, mg = milligram, mm = millimeter, mmHg = millimeters of mercury, mmol/L = millimole perliter, PC = principal component, Puncor = uncorrected p-values, PFDR = false-discovery rate-corrected p-values, RR = Blood pressure, sec = seconds.

*Presented as mean ± SD (N).

[†]Presented as χ2 for categorical and Cohen's d for continuous data.

[‡]Denotes statistical significance at FDR-corrected p<0.001

**Appendix 2—table 2.** Descriptive group statistics Hamburg City Health Study (HCHS).

| Metric* | Individuals with MetS | Matched controls | $P_{uncorr}$ | $P_{FDR}$ | Stat |
|---|---|---|---|---|---|
| Age (years) | 65.77±7.40 (759) | 65.97±7.52 (759) | 0.613 | 0.647 | 0.03 |
| Sex (% female) | 33.47 | 36.50 | 0.236 | 0.281 | 1.4 |
| Education (ISCED) | 2.37±0.58 (759) | 2.42±0.60 (759) | 0.09 | 0.114 | 0.09 |
| | | | | | |
| Metabolic syndrome criteria | | | | | |
| Waist circumference (cm) | 103.38±11.23 (754) | 91.45±11.36 (747) | <0.001 | <0.001† | −1.06 |
| RR$_{systolic}$ (mmHg) | 145.66±18.54 (740) | 140.17±21.10 (746) | <0.001 | <0.001† | −.28 |
| RR$_{diastolic}$ (mmHg) | 83.75±10.11 (740) | 82.00±10.40 (746) | 0.001 | 0.002 | −.17 |
| Antihypertensive therapy (%) | 52.60% | 26.22% | <0.001 | <0.001† | 108.89 |
| HDL (mg/dL) | 54.60±16.13 (751) | 67.63±17.46 (759) | <0.001 | <0.001† | 0.78 |
| Triglycerides (mg/dL) | 161.53±92.61 (751) | 91.23±30.62 (759) | <0.001 | <0.001† | −1.02 |
| Lipid lowering therapy (%) | 40.85% | 7.64% | <0.001 | <0.001† | 225.29 |
| Blood glucose (mg/dL) | 107.47±28.59 (742) | 90.99±10.87 (753) | <0.001 | <0.001† | −0.76 |
| Antidiabetic therapy (%) | 14.42% | 1.45% | <0.001 | <0.001† | 85.47 |
| | | | | | |
| Cognitive scores | | | | | |
| Animal Naming Test | 23.71±6.46 (712) | 24.77±6.75 (711) | 0.005 | 0.009 | 0.16 |
| Clock Drawing Test | 6.36±1.17 (730) | 6.39±1.16 (733) | 0.774 | 0.774 | 0.02 |
| Trail Making Test A (sec) | 41.26±14.28 (685) | 40.42±14.54 (675) | 0.321 | 0.359 | −0.06 |
| Trail Making Test B (sec) | 93.74±37.30 (675) | 89.89±37.69 (671) | 0.086 | 0.114 | −0.10 |
| Multiple-Choice Vocabulary Intelligence Test | 31.18±3.43 (603) | 31.71±3.22 (619) | 0.019 | 0.034 | 0.16 |
| Word List Recall | 7.42±1.89 (691) | 7.64±1.84 (673) | 0.057 | 0.083 | 0.12 |

*Appendix 2—table 2 continued on next page*

*Appendix 2—table 2 continued*

| Metric* | Individuals with MetS | Matched controls | $P_{uncorr}$ | $P_{FDR}$ | Stat |
|---|---|---|---|---|---|
| Imaging | | | | | |
| Mean cortical thickness (mm) | 2.327±0.08 (759) | 2.334±0.08 (757) | 0.045 | 0.071 | 0.1 |

Abbreviations: cm = centimeter, dL = deciliter, HDL = high-density lipoprotein, ISCED = International Standard Classification of Education, MetS = metabolic syndrome, mg = milligram, mm = millimeter, mmHg = millimeters of mercury, Puncor = uncorrected p-values, PFDR = false-discovery rate-corrected p-values, RR = Blood pressure, sec = seconds.

*Presented as mean ± SD (N).

†Denotes statistical significance at FDR-corrected p<0.001.

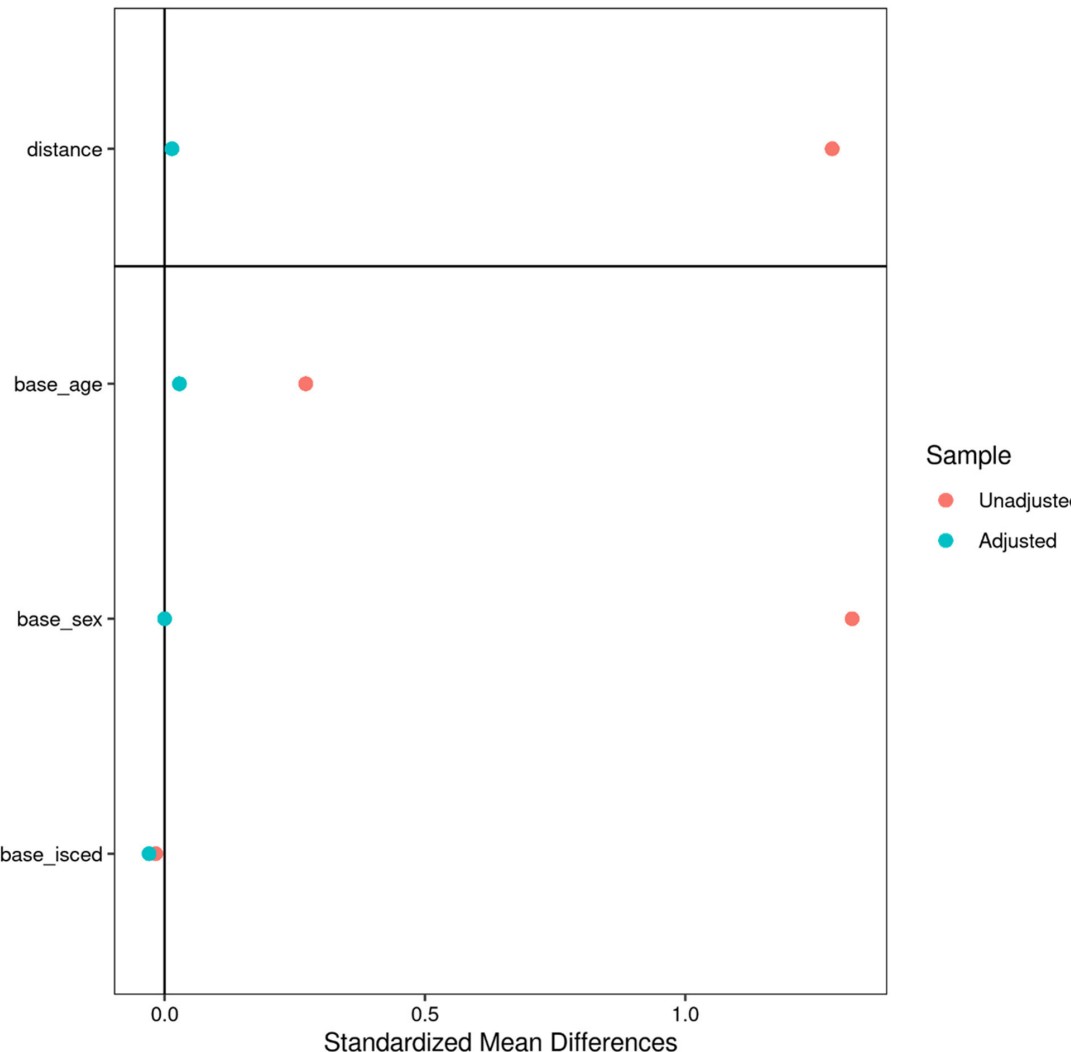

**Appendix 2—figure 1.** Matching - UK Biobank.

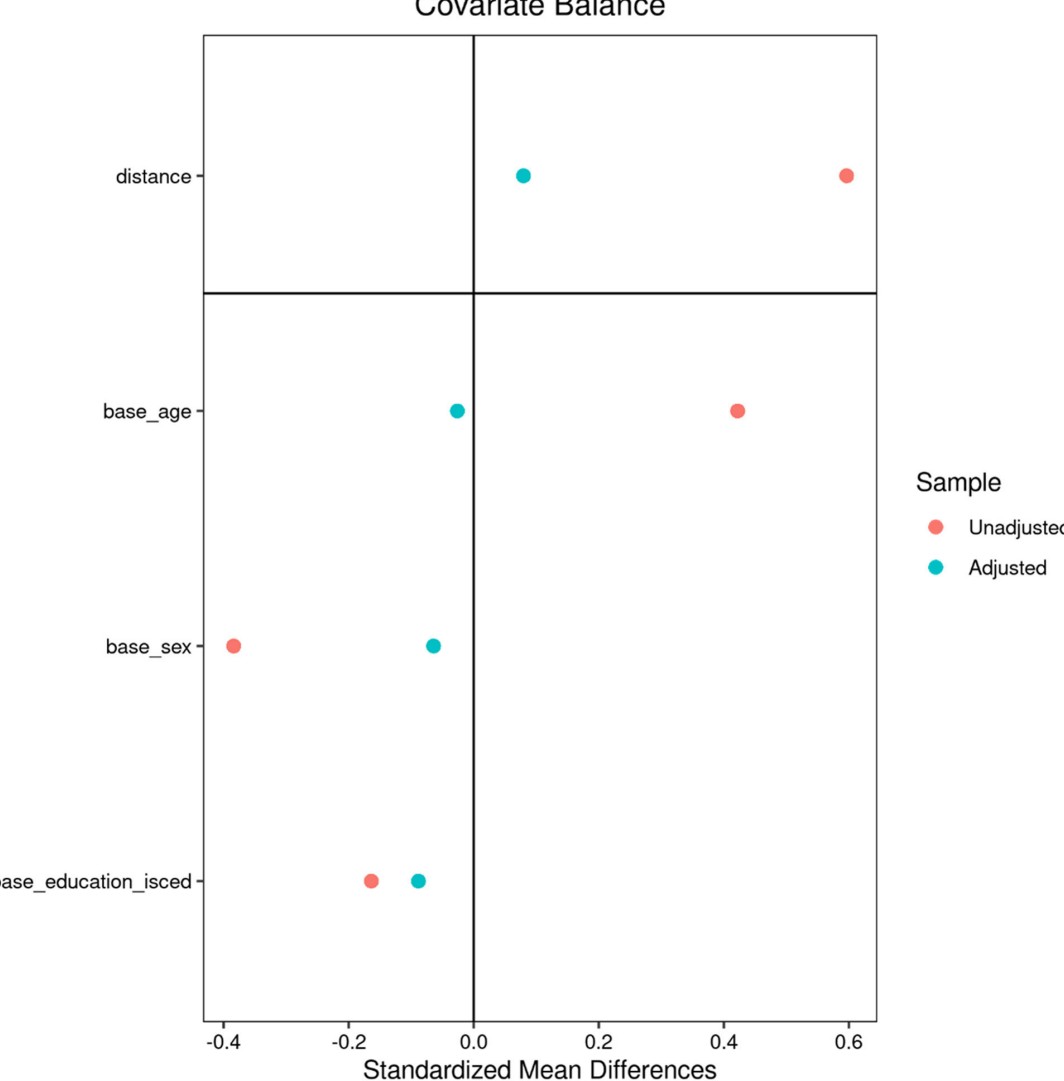

**Appendix 2—figure 2.** Matching – Hamburg City Health Study.

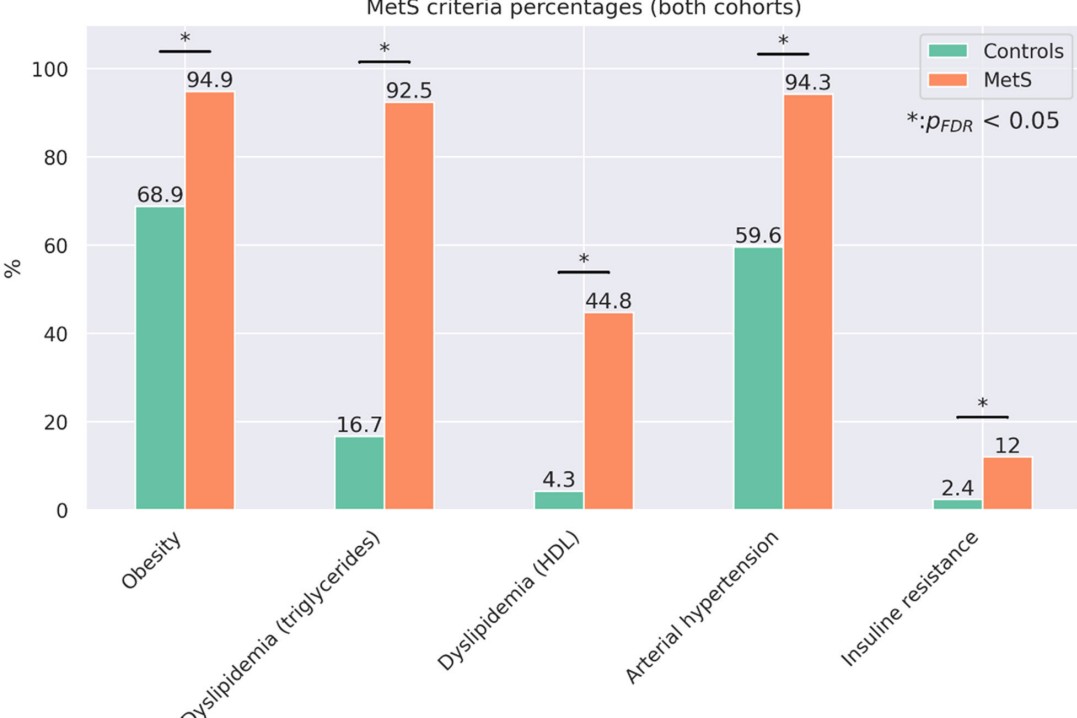

**Appendix 2—figure 3.** Proportion of metabolic syndrome criteria. Barplots indicate the percentage amount of metabolic syndrome (MetS) criteria that apply by group for the pooled sample. Significant group differences in $\chi$ 2-tests are highlighted by asterisks.

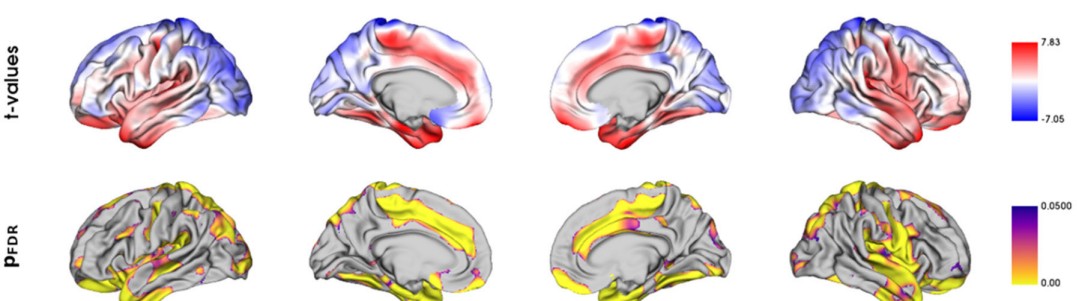

**Appendix 2—figure 4.** Vertix-wise group comparison of cortical thickness. Vertex-level group comparison between individuals with metabolic syndrome and matched controls. Resulting surface maps of standardized *t*-statistic estimates encode the group-differences between patients and controls, with lower cortical thickness in the metabolic syndrome (MetS) group being represented by a positive *t* and lower by a negative *t*. The vertex-wise *t*-statistic map was Schaefer-parcellated for the downstream analyses.

