## [Editor Report · eLife assessment]

This **important** work contributes to our understanding of the combined effects of metabolic syndrome on fronto-temporal gray matter morphology from two large-scale datasets. The evidence based on state-of-the art multivariate imaging analysis and detailed micro- and macrostructural contextualization analyses is **convincing** and provides an understanding of the neurological correlates of metabolic syndrome, although the study would have benefitted from the inclusion of longitudinal data.

---

## [Referee Report · Reviewer #1 (Public Review)]

Summary:

In their study, Petersen et al. investigated the relationship between parameters of metabolic syndrome (MetS) and cortical thickness using partial least-squares correlation analysis (PLS) and performed subsequently a group comparison (sensitivity analysis). To do this, they utilized data from two large-scale population-based cohorts: the UK BioBank (UKB) and the Hamburg City Health Study (HCHS). They identified a latent variable that explained 77% of the shared variance, driven by several measures related to MetS, with obesity-related measures having the strongest contribution. Their results highlighted that higher cortical thickness in the orbitofrontal, lateral prefrontal, insular, anterior cingulate, and temporal areas is associated with lower MetS severity. Conversely, the opposite pattern was observed in the superior frontal, parietal, and occipital regions. A similar pattern was then observed in the sensitivity analysis when comparing two groups (MetS vs. matched controls) separately.

Interestingly, after including HbA1c (a blood glycemic marker, which reflects insulin resistance much better than non-fasting glucose) in their revision, the authors identified a second latent variable accounting for 22% of shared variance mostly driven by HbA1c and blood glucose. The authors conclude that the distinct covariance profile of this variable likely indicates a separate pathological mechanistic connection between MetS components and brain morphology.

They then mapped local cellular and network topological attributes to the observed cortical changes associated with MetS. This was achieved using cell-type-specific gene expressions from the Allen Human Brain Atlas and the group consensus functional and structural connectomes of the Human Connectome Project (HCP), respectively. This contextualization analysis allowed them to identify potential cellular contributions in these structures driven by endothelial cells, microglial cells, and excitatory neurons. It also indicated functional and structural interconnectedness of areas experiencing similar MetS effects.

Strengths:

The effects of metabolic syndrome on the brain are still incompletely understood, and such multi-scale analyses are important for the field. Despite the study's sole 'correlation-based' nature, it yields valuable results, including several scales of brain parameters (cortical thickness, cellular, and network-based). The results are robust and benefit from two 'large-scale' datasets, resulting in highly powered statistics

Weaknesses:

The weakness of this study lies mostly on the non-causative approach used here. Nevertheless, the authors are aware of the limitations of the study and carefully frame their language accordingly.

---

## [Referee Report · Reviewer #2 (Public Review)]

Summary:

In this manuscript, Petersen et al. aimed for a comprehensive assessment of the relationship between cardiometabolic risk factors and cortical thickness. They found that a latent variable reflecting higher obesity, hypertension, LDL cholesterol, triglyerides, non-fasting glucose, HbA1c and lower HDL cholesterol was associated with lower cortical thickness in orbitofrontal, lateral prefrontal, insular, anterior cingulate and temporal areas as well as lower subcortical volumes. In sensitivity analyses they showed that this pattern replicated across cohorts and was also consistent with a clinical definition of the metabolic syndrome.

Further, when including cognition into the multivariate analysis, the pattern remained unchanged and mediation analyses showed that the relationship between the first latent variable and worse cognitive performance across several tests was mediated by the brain morphological differences.

The authors investigated the cell types implicated in the regions associated with cardiometabolic risk using the Allen Brain Atlas and found that the density of excitatory neurons type 8, endothelial cells and microglia reliably co-located with the pattern of cortical thickness. Furthermore, they showed that cortial regions more strongly associated with MetS were more closely structurally & functionally connected than others.

Strengths:

This study performed a comprehensive assessment of the combined association of cardiometabolic risk factors and brain structure and investigated micro-and macroscopic underpinnings. A major strength of the study is the methodological approach of partial least squares which allows one to not single out risk factors but to take them into account simultaneously. The large sample size from two cohorts allowed for different sensitivity analyses and convincing evidence for the stability of the first latent variable. The authors demonstrated that the component was also reliably related to cognitive performance and that the association of the individual cardiometabolic risk on cognition was mediated by brain morphological differences, replicating multiple previous studies which evidenced associations of different components of the MetS with worse cognitive performance.

The novel contribution of the study lies in the virtual histology and brain topology investigation of the cortical pattern related to MetS. The virtual histology provided convincing evidence of the co-localization of endothelial, glial and excitatory neuronal cells with the regions of MetS-associated cortical thinning while the brain topology analysis highlighted the disproportionate structural and functional connectivity between associated regions. This analysis provides insights into the role of inflammatory processes and the intricate link between gray matter morphology and microvasculature, both locally and in relation to long-range connectivity. This information is valuable to inform future mechanistic studies.

Weaknesses:

The study is exclusively cross-sectional which does not allow disentangling potential causes from consequences. While studies indicate that most of the differences seen in middle age are probably consequences of the MetS on the vasculature, blood-brain barrier or inflammatory processes, differences in cortical morphology might also represent a risk factor for weight gain.

The study is exploratory in nature and for the contextualization analyses it is difficult to judge whether those were selected from a larger pool of analyses.

---

## [Author Response]

The following is the authors’ response to the original reviews.

**Reviewer 1**
Comment 1.1: “Did the UKB or HCHS datasets have information on accurate markers of insulin resistance, such as HbA1c or HOMA-IR (if fasting glucose was not available)? Looking at that data would allow us to determine the contribution of insulin resistance to the observed cortical phenotype.”

Reply 1.1: We appreciate the insightful suggestion from the reviewer. In response, we incorporated the HbA1c into our analysis, enhancing its sensitivity to potential effects of insulin resistance. Subsequently, our analysis was reperformed, integrating HbA1c alongside non-fasting blood glucose in the PLS. This addition did not alter our main results, i.e., that of the PLS, virtual histology, and network contextualization analysis. Notably, as a result of the inclusion of HbA1c, the second latent variable now accounted for a greater shared variance (22.13%), with HbA1c showing the highest loading among MetS component variables. The manuscript has been thoroughly revised to incorporate these results.

Comments 1.2: “(Results, p.13, 291-292) "A correlation matrix relating all considered MetS component measures is displayed in supplementary figure S12. Please clarify in this figure labels whether this was non-fasting glucose. If this is non-fasting glucose, it is not a MetS-related risk factor. The reader might be misled into thinking that fasting-glucose has a weak correlation, while its contribution (and the effect of insulin resistance) was not studied here.”“Table S8 and Table S9: Is the glucose metric here measured following fasting? If not, this should not be listed as a metabolic syndrome criterion. Or it should be specified that it isn't fasted glucose, otherwise, it sounds misleading.”

Reply 1.2: We thank the reviewer for bringing this ambiguity to our attention. The initial analysis included only non-fasting plasma glucose in the PLS, as fasting plasma glucose data was unavailable for UKB and HCHS participants. Following your suggestion in reply 1.1, we have now incorporated HbA1c, a more indicative marker of insulin resistance. We retained non-fasting blood glucose in our analysis, recognizing its relevance as a diagnostic variable for type 2 diabetes mellitus, although it is less informative than fasting plasma glucose, HbA1c, or HOMA-IR. This decision is substantiated by the significant correlation found between non-fasting plasma glucose and HbA1c in our sample (r=.49).

To enhance clarity, we have revised the methods section to explicitly mention that the study investigates non-fasting blood glucose. The revised sentence reads: “Here, we related regional cortical thickness and subcortical volumes to clinical measurements of MetS components, i.e., obesity (waist circumference, hip circumference, waist-hip ratio, body mass index), arterial hypertension (systolic blood pressure, diastolic blood pressure), dyslipidemia (high density lipoprotein, low density lipoprotein, total cholesterol, triglycerides) and insulin resistance (HbA1c, non-fasting blood glucose).”

Additionally, we have updated the caption of supplementary figure S13 (formerly supplementary figure S12) to clearly indicate the investigation of non-fasting plasma glucose. The table detailing diagnostic MetS criteria (supplementary table S2) has also been amended to clarify the absence of fasting plasma glucose data in our study and to indicate that only data on antidiabetic therapy and diagnosis of type 2 diabetes mellitus were used as criteria for insulin resistance in the case-control analysis.

Comment 1.3: “I do not understand how the authors can claim there is a deterministic relationship there if all the results are only correlational or comparative. Can the differences in functional connectivity and white matter fiber tracts observed not be caused by the changes in cortices they relate to? How can the authors be sure the network organisation is shaping the cortical effects and not the opposite (the cortical changes influence the network organisation)? This should be further discussed or explained.”

Reply 1.3: We agree with the reviewer's comment on the non-causative nature of our data and have accordingly revised the discussion section to reflect a more cautious interpretation of our findings. We have carefully reframed our language to avoid any implications of causality, ensuring the narrative aligns with the correlational nature of our data. Nevertheless, we believe that exploring causal interpretations can offer valuable clinical insights. Therefore, while moderating our language, we have maintained certain speculative discussions regarding potential causative pathomechanistic pathways.

Comment 1.4: “The hippocampus is also an area where changes have consistently been observed. Why did the authors limit their analysis to the cortex.”

Reply 1.4: We appreciate this reviewer comment. In response, we have added volumes of Melbourne Subcortical Atlas parcels (including the hippocampus) to the analysis. Corresponding results are now shown in figure 2. The subcortical bootstrap ratios indicated that higher MetS severity was related to lower volumes across all investigated subcortical structures.

Comment 1.5: “Which field ID of the UK biobank are the measures referring to? If possible, please specify the Field ID for each of the UKB metrics used in the study.”

Reply 1.5: We thank the reviewer for the recommendation. The Field IDs used in our study are now listed in supplementary figure S1.

Comment 1.6: “Several Figures were wrongly annotated, making it hard to follow the text.”

Reply 1.6: Thank you for bringing the annotation issues to our awareness. We have thoroughly edited all annotations which should now correctly reference the figure content.

**Reviewer 2**
Comment 2.1: “Do the authors have the chance to see how the pattern relates to changes in cognitive function in the UKBB and possibly HCHS? This could help to provide some evidence about the directionality of the effect.”Reply 2.1: Thank you for your suggestion. We acknowledge the potential value of investigating graymatter morphometric data alongside longitudinal information on cognitive function. Although we concur with the significance of this approach, we are constrained by the ongoing processing of the UKB's imaging follow-up data and the pending release of the HCHS follow-up data. Consequently, our current analysis cannot incorporate this aspect for now. We plan to explore the relationship between MetS, cognition and brain morphology using longitudinal data as soon as it becomes available.Comment 2.2: “Also, you could project new data onto the component and establish a link with cognition in a third sample which would be even more convincing. I can offer LIFE-Adult study for this aim.”

Reply 2.2: We are grateful for your recommendation to enhance our study's robustness by including a third sample to establish a cognitive link. While we recognize the merit of such a sensitivity analysis, we believe that our current dataset, derived from two large, independent cohorts, is sufficiently comprehensive for the scope of our current analysis. However, we are open to considering this approach in future studies and appreciate your offer of the LIFE-Adult study. We would welcome further conversation with you regarding future joint projects.

Comment 2.3: “The sentences (p.17, ll.435 ff) seem to repeat: "Interestingly, we also observed a positive relationship between cortical thickness and MetS in the superior frontal, parietal and occipital lobe. Interpretation of this result is, however, less intuitive. We also noted a positive MetS-cortical thickness association in superior frontal, parietal and occipital lobes, a less intuitive finding that has been previously reported [60,61].”

Reply 2.3: Thank you for making us aware of this duplication. We have deleted the first part of the section. It now reads “We also noted a positive MetS-cortical thickness association in superior frontal, parietal and occipital lobes, a less intuitive finding that has been previously reported.”

Comment 2.4: “I would highly appreciate empirical evidence for the claim in ll. 442 "In support of this hypothesis, the determined cortical thickness abnormality pattern is consistent with the atrophy pattern found in vascular mild cognitive impairment and vascular dementia" Considering the previous reports about the co-localization of obesity-associated atrophy and AD neurodegeneration (Morys et al. 2023, DOI: 10.3233/JAD-220535), that most dementias are mixed and that MetS probably increases dementia risk through both AD and vascular mechanisms, I feel such "binary" claims on VaD/AD-related atrophy patterns should be backed up empirically.”

Reply 2.4: Thank you for highlighting the need for clarity in differentiating between vascular and Alzheimer's dementia. We recognize the intricate overlap in dementia pathologies. Acknowledging the prevalence of mixed dementia and the influence of MetS on both AD and vascular mechanisms, we realize our original statement might have implied a specificity to vascular dementia, which was not intended.

To address your concern, we have revised our statement to avoid an exclusive focus on vascular pathology, ensuring a more balanced representation of dementia types. Additionally, we have included Morys et al. 2023 as a reference. The section now reads: “In support of this hypothesis, the determined brain morphological abnormality pattern is consistent with the atrophy pattern found in vascular mild cognitive impairment, vascular dementia and Alzheimer’s dementia.”

Comment 2.5: “I wonder how specific the cell-type results are to this covariance pattern. Maybe patterns of CT (independent of MetS) show similar associations with one or more of the reported celltypes? Would it be possible to additionally show the association of the first three components of general cortical thickness variation with the cell type densities?”

Reply 2.5: Thank you for your query regarding the specificity of the cell-type results to the observed covariance pattern. To address this, we have conducted a virtual histology analysis of the first three latent variables of the main analysis PLS. The findings of this extended analysis have been detailed in the supplementary Figure S21. The imaging covariance profile of latent variable 2 was significantly associated with the density of excitatory neurons of subtype 3. The imaging covariance profile linked to latent variable 3 showed no significant association of cell type densities. Possibly, latent variable 3 represents only a noise component as it explained only 2.12% of shared variance. We hope this addition provides a clearer understanding of the specificity of our main results.

Comment 2.6: “I agree that this multivariate approach can contribute to a more holistic understanding, yet I would like to see the discussion expanded on how to move on from here. Should we target the MetS more comprehensively or would it be best to focus on obesity (being the strongest contributor and risk factor for other "downstream" conditions such as T2DM)? A holistic approach is somewhat at odds with the in-depth investigation of specific mechanisms.”

Reply 2.6: We value your suggestion to elaborate on the implications of our findings. Our study indicates that obesity may have the most pronounced impact on brain morphology among MetS components, suggesting it as a key contributor to the clinical-anatomical covariance pattern observed in our analysis. This highlights obesity as a primary target for future research and preventive strategies. However, we believe that our results warrant further validation, ideally through longitudinal studies, before drawing definitive clinical conclusions.

Additionally, our study endorses a comprehensive approach to MetS, highlighting the importance of considering the syndrome as a whole to gain broader insights. We want to clarify, however, that such an approach is meant to complement, rather than replace, the study of individual cardiometabolic risk factors. The broad perspective our study adopts is facilitated by its epidemiological nature, which may not be as applicable in experimental settings that are vital for deriving mechanistic disease insights.

To reflect these points, we have expanded the discussion in our manuscript to include a more detailed consideration of these implications and future research directions.

Comment 2.7: “Please report the number of missing variables.”

Reply 2.7: Thank you for your request to report the number of missing variables. We would like to direct your attention to table 1, where we have listed the number of available values for each variable in parentheses. To determine the number of missing variables, one can subtract these numbers from the total sample size.

Comment 2.8: “Was the pattern similar in pre-clinical (pre-diabetes, pre-hypertension) vs. clinical conditions?“

Reply 2.8: Thank you for your interest in the applicability of our findings across different MetS severity levels. Our analysis employs a continuous framework to encompass the entire range of vascular and cardiometabolic risks, including those only mildly affected by MetS. The linear relationship we observed between MetS severity and gray matter morphology patterns, as illustrated in Figure 2d, supports the interpretation that our findings apply to the entire spectrum of MetS severities.

Comment 2.9: “How did you deal with medication (anti-hypertensive, anti-diabetic, statins..)?”

Reply 2.9: Information on medication was considered for defining MetS for the case-control sensitivity analysis but was not included in the PLS. Detailed information can be found in table 1.

Comment 2.10: “It would be really interesting to determine the genetic variations associated with the latent component. Have you considered doing a GWAS on this, potentially in the CHARGE consortium or with UKBB as discovery and HCHS as replication sample?”

Reply 2.10: Thank you for your valuable suggestion regarding the implementation of a GWAS. We agree that incorporating a GWAS would provide significant insights, but we also recognize that it extends beyond the scope of our current analysis. However, we are actively planning a follow-up analysis. This subsequent analysis will encompass a comprehensive examination of both genetic variation and imaging findings in the context of MetS.

Comment 2.11: “Please provide more information on which data fields from UKBB were used exactly (e.g. in github repository).”

Reply 2.11: We appreciate your recommendation. The details regarding the Field IDs used in our study have been included as supplementary table S1.

**Reviewer 3**
Comments 3.1: “After a thorough review of the methods and results sections, I found no direct or strong evidence supporting the authors' claim that the identified latent variables were related to more severe MetS to worse cognitive performance. While a sub-group comparison was conducted, it didnot adequately account for confounding factors such as educational level.”“Page 18-19 lines 431-446: the fifth paragraph in the discussion section. - As previously mentioned in the "Weaknesses" section, this study did not conduct a direct association analysis between MetS and cognitive levels without considering subgroup comparisons. Hence, I recommend the content of this paragraph warrants careful reconsideration.”

Reply 3.1: We acknowledge the reviewer's constructive feedback regarding our analysis of cognitive data. We have performed a mediation analysis relating the subject-specific clinical PLS score of latent variable 1 representing MetS severity and cognitive test performances and testing for mediating effects of the imaging PLS score capturing the MetS-related brain morphological abnormalities. The imaging score was found to statistically mediate the relationship between the clinical PLS score and executive function and processing speed, memory, and reasoning test performance. These findings highlight brain structural differences as a relevant pathomechanistic correlate in the relationship of MetS and cognition. Corresponding information can now be found in figure 3, methods section 2.6.2, result section 3.3 and discussion section 4.2.

Moreover, we would like to apologize for any confusion caused by previous unclear presentation. Our study further incorporates association analyses between MetS, brain structure, and cognition using MetS components, regional brain morphological measures, and cognitive performance data in a PLS to investigate whether cognitive measures contribute to the latent variable. These analyses were separately performed on the UK Biobank and HCHS datasets, due to their distinct cognitive assessments. We adjusted for age, sex, and education in the subgroup analyses by removing their effects from the input variables. These relationships are detailed in supplementary figures S16b and S17b, with loadings close to zero for age, sex, and education, confirming effective deconfounding.

In sum, we greatly appreciate the suggestion to conduct a mediation analysis, which has substantially enhanced the strength and relevance of our analysis.

Comment 3.2: “I would suggest the authors provide a more comprehensive description of the metrics used to assess each MetS component, such as obesity (incorporating parameters like waist circumference, hip circumference, waist-hip ratio, and body mass index) and arterial hypertension (detailing metrics like systolic and diastolic blood pressure), etc.”

Reply 3.2: Thank you for your suggestion regarding a more detailed description of the metrics for assessing each component of MetS. We would like to point out that the specific metrics used, including those for obesity (such as waist circumference, hip circumference, waist-hip ratio, and body mass index) and arterial hypertension (including systolic and diastolic blood pressure), are comprehensively detailed in table 1 of our manuscript. We hope this table provides the clarity and specificity you are seeking regarding the MetS assessment metrics in our study.

Comment 3.3: “I recommend the inclusion of an additional, detailed flowchart to further illustrate the procedure of virtual histology analysis. This would enhance the clarity of the methodological approach and assist readers in better comprehending the analysis method.”

Reply 3.3: Thank you for your suggestion. Recognizing the challenges in visually representing many of our analysis steps, we have instead supplemented our manuscript with additional references. These references provide a clearer understanding of our virtual histology approach, particularly focusing on the processing of regional microarray expression data.

The corresponding sentence reads: “Further details on the processing steps covered by ABAnnotate can be found elsewhere (https://osf.io/gcxun) [42]”

Comment 3.4: “Why were both brain hemispheres used instead of solely utilizing the left hemisphere as the atlas, especially considering that the Allen Human Brain Atlas (AHBA) only includes gene data for the right hemisphere for two subjects?”

Reply 3.4: Thank you for your query regarding our decision to use both brain hemispheres instead of solely the left hemisphere, especially considering the Allen Human Brain Atlas (AHBA) predominantly featuring gene data from the left hemisphere. Given the AHBA's limited spatial coverage of expression data in the right hemisphere, our approach involved mirroring the existing tissue samples across the left-right hemisphere boundary using the abagen toolbox,1 a practice supported by findings that suggest minimal lateralization of microarray expression.2,3 Further details are provided in previous work employing ABAnnotate.4 These studies are now referenced in our methods section.

Comment 3.5: “The second latent variable was not further discussed. If this result is deemed significant, it warrants a more detailed discussion.”

Reply 3.5: Thank you for the suggestion. We have added a paragraph to the discussion that discusses the second latent variable in greater detail. It reads: “The second latent variable accounted for 22.33% of shared variance and linked higher insulin resistance and lower dyslipidemia to lower thickness and volume in lateral frontal, posterior temporal, parietal and occipital regions. The distinct covariance profile of this latent variable, compared to the first, likely indicates a separate pathomechanistic connection between MetS components and brain morphology. Given that HbA1c and blood glucose were the most significant contributors to this variable, insulin resistance might drive the observed clinicalanatomical relationship.”

Comment 3.6: “I suggest appending positive MetS effects after "..., insular, cingulate and temporal cortices;" for two reasons: (a). The "positive MetS effects" might represent crucial findings that should not be omitted. (b). Including both negative and positive effects ensures that subsequent references to "this pattern" are more precise.”

Reply 3.6: We concur with the notion that the positive MetS effects should be highlighted as well. We modified the first discussion paragraph now mentioning them.

Comment 3.7: “I would appreciate further clarification on this sentence and the use of the term "uniform" in this context. Does this suggest that despite the heterogeneity in the physiological and pathological characteristics of the various MetS components (e.g., obesity, hypertension), their impacts on cortical thickness manifest similarly? How is it that these diverse components lead to "uniform" effects on cortical thickness? Does this observation align with or deviate from previous findings in the literature?”

Reply 3.7: Thank you for highlighting the ambiguity in our previous explanation. We agree that the complexity of the relationship between MetS components and brain morphology requires clearer articulation. To address this, we have revised the relevant sentence for better clarity. It now reads: „This finding indicates a relatively uniform connection between MetS and brain morphology, implying that the associative effects of various MetS components on brain structure are comparatively similar, despite the distinct pathomechanisms each component entails.“

Comment 3.8: “Figure 1 does not have the labels "(c)" and "(d)". ”

Reply 3.8: Thank you. We have modified figure 1 and made sure that the caption correctly references its content.

Comment 3.10: “Incorrect figure/table citation:Page 18 line 418: "(figure 2b and 1c)" à (figure 2b and 2c).Page 18 line 419: "(supplementary figures S8 and S12-13)" à (supplementary figures S11 and S1516).In the supplementary material, "Text S5 - Case-control analysis" section contains several figure or table citation errors. Please take a moment to review and correct them.”

Reply 3.10: Thank you for bringing this to our attention. We have corrected the figure and table citation errors.

Comment 3.11: “Page 8 line 184: The more commonly used term is "insulin resistance" rather than "insuline resistance.”

Reply 3.11: We now use “insulin resistance” throughout the manuscript.

Comment 3.12: “Nevertheless, variations in gene sets may introduce a degree of heterogeneity in the results (Seidlitz, et al., 2020; Martins et al., 2021). Consequently, further validation or exploratory analyses utilizing different gene sets can yield more compelling results and conclusions.”

Reply 3.12: Thank you for your insightful comment regarding the potential heterogeneity introduced by variations in gene sets. We agree that exploring different gene sets could indeed enhance the robustness and generalizability of our findings. However, we think conducting a comprehensive methodological analysis of the available cell-type specific gene sets is a substantial effort and warrants its own investigation to thoroughly implement it and assess its implications. We also like to highlight that we are adhering to previous practices in our analysis setup.4,5

References

(1) Markello RD, Arnatkeviciute A, Poline JB, Fulcher BD, Fornito A, Misic B. Standardizing workflows in imaging transcriptomics with the abagen toolbox. Jbabdi S, Makin TR, Jbabdi S, Burt J, Hawrylycz MJ, eds. eLife. 2021;10:e72129. doi:10.7554/eLife.72129

(2) Hawrylycz MJ, Lein ES, Guillozet-Bongaarts AL, et al. An anatomically comprehensive atlas of the adult human brain transcriptome. Nature. 2012;489(7416):391-399. doi:10.1038/nature11405

(3) Hawrylycz M, Miller JA, Menon V, et al. Canonical genetic signatures of the adult human brain. Nat Neurosci. 2015;18(12):1832-1844. doi:10.1038/nn.4171

(4) Lotter LD, Saberi A, Hansen JY, et al. Human cortex development is shaped by molecular and cellular brain systems. Published online May 5, 2023:2023.05.05.539537. doi:10.1101/2023.05.05.539537

(5) Lotter LD, Kohl SH, Gerloff C, et al. Revealing the neurobiology underlying interpersonal neural synchronization with multimodal data fusion. Neuroscience & Biobehavioral Reviews.2023;146:105042. doi:10.1016/j.neubiorev.2023.105042